# Cryo-EM analyses unveil details of mechanism and targocil-II mediated inhibition of *S. aureus* WTA transporter TarGH

Franco K. K. Li [1,4], Shaun C. Peters [1,4], Liam J. Worrall[1,2,4], Tianjun Sun[1], Jinhong Hu[1], Marija Vuckovic[1], Maya Farha[3], Armando Palacios[1], Nathanael A. Caveney [1,2], Eric D. Brown [3] & Natalie C. J. Strynadka [1,2] ✉

Wall teichoic acid (WTA) is a polyol phosphate polymer that covalently decorates peptidoglycan of gram-positive bacteria, including *Staphylococcus aureus*. Central to WTA biosynthesis is flipping of lipid-linked precursors across the cell membrane by TarGH, a type V ABC transporter. Here, we present cryo-EM structures of *S. aureus* TarGH in the presence of targocil-II, a promising small-molecule lead with β-lactam antibiotic synergistic action. Targocil-II binds to the extracellular dimerisation interface of TarG, we suggest mimicking flipped but not yet released substrate. In absence of targocil-II and in complex with ATP analogue ATPγS, determined at 2.3 Å resolution, the ATPase active site is allosterically inhibited. This is due to a so far undescribed D-loop conformation, potentially minimizing spurious ATP hydrolysis in the absence of substrate. Targocil-II binding comparatively causes local and remote conformational changes through to the TarH active site, with the D-loop now optimal for ATP hydrolysis. These structures suggest an ability to modulate ATP hydrolysis in a WTA substrate dependent manner and a jammed ATPase cycle as the basis of the observed inhibition by targocil-II. The molecular insights provide an unprecedented basis for development of TarGH targeted therapeutics for treatment of multidrug-resistant *S. aureus* and other gram-positive bacterial infections.

Antimicrobial resistance (AMR) is a leading global health crisis, with 10-million AMR related annual deaths expected by 2050[1]. The World Health Organization (WHO) has designated ESKAPEE[2] bacterial pathogens (*Enterococcus faecium, Staphylococcus aureus, Klebsiella pneumoniae, Acinetobacter baumannii, Pseudomonas aeruginosa, Enterobacter spp. and Escherichia coli*) as amongst the most critical targets for the development of therapeutics. *S. aureus* is a common nosocomial pathogen and is associated with hospital-acquired sepsis, bacteremia and endocarditis[3]. Effective treatments for *S. aureus* infection, including tetracycline (dysregulation of protein synthesis), ciproflaxin (inhibition of nucleic acid synthesis), or β-lactams (prevention of essential cell wall cross-linking) are complicated by multi-

[1]Department of Biochemistry and Molecular Biology and the Centre for Blood Research, University of British Columbia, Vancouver, BC, Canada. [2]High Resolution Macromolecular Cryo-Electron Microscopy (HRMEM) Facility, University of British Columbia, Vancouver, BC, Canada. [3]Department of Biochemistry and Biomedical Sciences, McMaster University, Hamilton, ON, Canada. [4]These authors contributed equally: Franco K. K. Li, Shaun C. Peters, Liam J. Worrall. ✉e-mail: ncjs@mail.ubc.ca

drug resistance[4–6]. Notably, methicillin-resistant (MRSA) and vancomycin-resistant *S. aureus*, pose a major clinical and economic burden and exemplify the need for new antibiotics targeting other enzyme activities including the largely untapped biogenesis of the unique to bacteria cell wall ultrastructure.

A defining feature of *S. aureus* and other gram-positive bacteria is a single membrane bilayer surrounded by a thick matrix of pepti-doglycan (PG), a mesh of long polymers of repeating N-acetylglucosamine and N-acetylmuramic acid (GlcNAc–MurNAc) crosslinked via covalent peptide linkages[7]. In many gram-positive bacteria, PG is further decorated by polymerised phosphodiester-linked polyols known as wall teichoic acid (WTA)[8], which can con-stitute up to 60% of the cell wall mass[9,10]. WTA plays a critical role in cell division[11], localising enzymes for division-mediated PG degradation[12] and modulating the activity of PG-synthetic enzymes for regulation of PG crosslinking[13]. WTA also plays a role in pathogenesis including host colonisation[14], biofilm formation[15] and immune evasion[16–18]. Impor-tantly, WTA is central to β-lactam resistance in MRSA and inhibiting its

synthesis sensitises MRSA to this important class of antibiotics[19,20], underpinning the rationale for the development of inhibitors against WTA biosynthesis.

The WTA biosynthetic pathway is comprised of a number of sequential enzymatic steps that catalyse the polymerisation of a WTA precursor on an undecaprenyl-pyrophosphate lipid carrier localised within the inner leaflet of the cytoplasmic membrane ($C_{55}$-PP-GlcNAc-ManNAc-GroP-(RboP)$_{n<40}$; Fig. 1a). The lipid-linked product must be flipped to the extracellular leaflet for subsequent glycosyl transfer and covalent attachment of the glycan polymer to MurNAc moieties of PG[21]. This flipping is carried out by the ATP-binding cassette (ABC) transporter TarGH (Fig. 1b). TarGH is a heterotetramer composed of two TarG polytopic transmembrane domains (TMD) and two TarH nucleotide binding domains (NBD). Like other ABC transporters, TarGH uses the energy derived from the ATP hydrolysis cycle in the NBDs to drive substrate transport across the membrane via the TMDs.

TarGH is non-redundant in *S. aureus*, with deletion of *tarG* or *tarH* genes resulting in non-viable phenotypes through presumed toxic

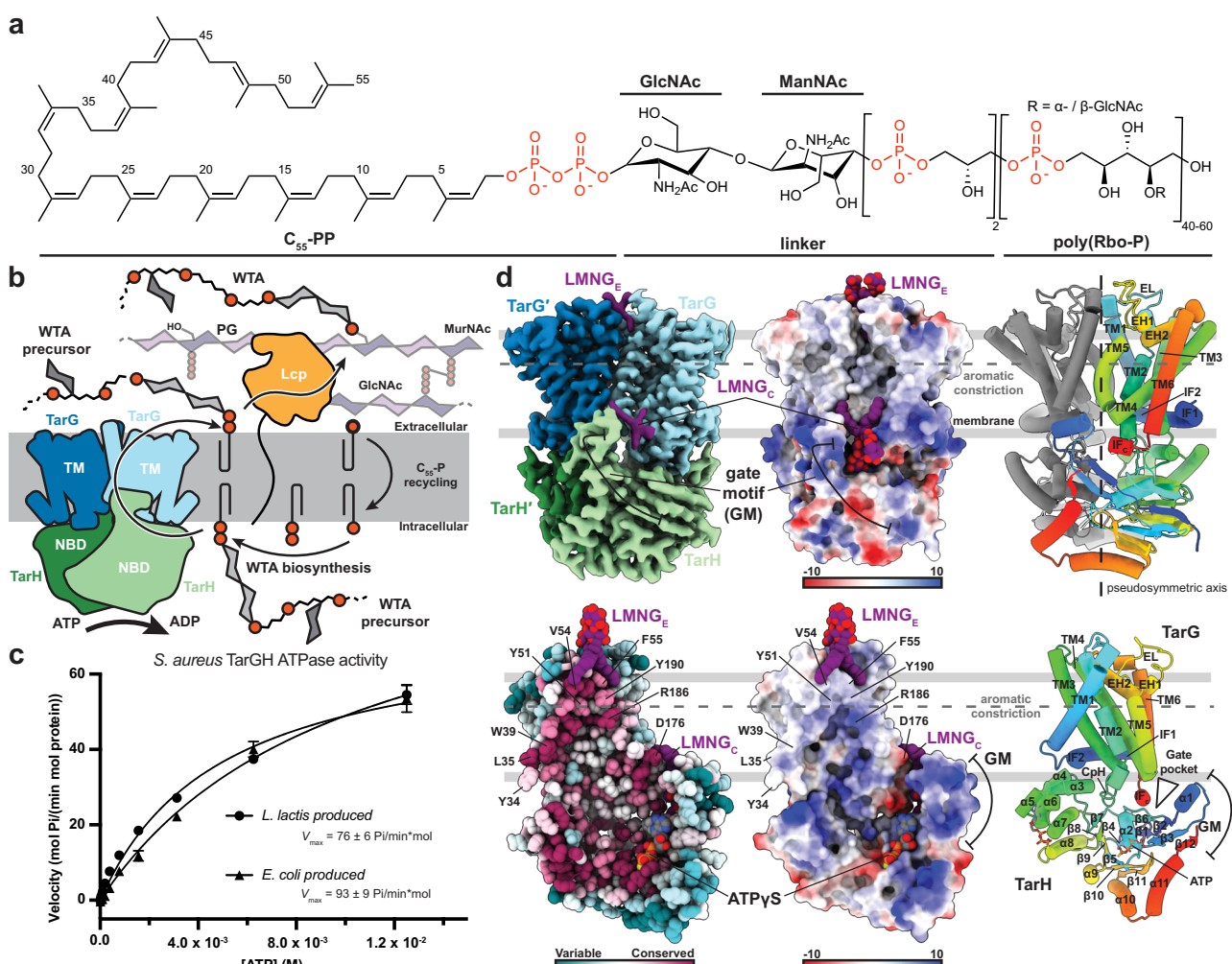

**Fig. 1 | Activity and general structural features of *S. aureus* TarGH. a** 2D chemical structure of the lipid-linked WTA precursor substrate of *S. aureus* TarGH. **b** Cartoon schematic of WTA precursor transport, peptidoglycan attachment and $C_{55}$-P recy-cling within *S. aureus*. **c** ATPase activity of *S. aureus* TarGH expressed in *E. coli* or *L. lactis* measured using a malachite green assay. Data points represent the mean of three independent samples ± the standard deviation while *V*max error is reported within the 95% confidence interval. **d** Cryo-EM reconstruction of *E. coli* expressed *S. aureus* TarGH (2.3 Å resolution) bound to ATPγS viewed as heterotetrameric (top) or heterodimeric half-transporter (bottom) assemblies. Membrane spanning TarG

and nucleotide binding domain TarH dimers are coloured light/dark blue and green (top, left), by electrostatic potential surface depiction (middle), sequence con-servation (Consurf) (bottom, left), or a cartoon depiction with one monomer of each coloured rainbow from N- (blue) to C- (red) terminus (right). Bound LMNG and ATPγS molecules are shown as spheres with heteroatom colouring (carbons purple or grey, respectively) and transmembrane (TM) helices, reentrant helices (EH), interfacial helices (IF1/2) and extracellular loop (EL) of TarG are labelled along with the gate motif (GM) and gate helix (α1) of TarH. Vertical disposition of the inter-TarG aromatic constriction is denoted by a grey dashed line.

accumulation of lipidated WTA intermediates in the membrane[22]. TarG is also accessible on the bacterial cell surface making it a compelling target for drug development[23–27], as validated by the identification of two TarGH inhibitors, targocil[23] and targocil-II[24]. These compounds are bacteriostatic in nature[26] and, importantly, act in synergy with β-lactams[27–29]. Despite these promising leads, high resolution structural information important for understanding mechanism and facilitating further drug discovery in clinically relevant TarGH homologues is lacking. A single structural state of non-pathogenic *Alicyclobacillus herbarius* TarGH (~50% sequence identity with *S. aureus*) has recently been determined by cryo electron microscopy (cryo-EM) methods[30]. We note in that structure, however, that the challenges associated with model building at moderate global resolution (~3.9 Å) and poorer local resolution of the NBDs resulted in the misidentification of a nucleotide-free state, with the deposited map showing density supporting bound ATP. Thus, many questions of TarGH action including substrate binding, inhibition and species-specific details of the *S. aureus* pathogenic variants remain unanswered.

In this study, we present cryo-EM structures of *S. aureus* TarGH at resolutions from 2.3 to 3.0 Å, with captured states varying in the presence of ATP analogues, ordered disaccharide detergent lauryl maltose neopentyl glycol (LMNG) and targocil-II. Together these structures illuminate experimentally determined details of not only targocil-II binding, but as well an observed series of conformational changes that span from extracellular to cytosolic nucleotide binding domains that point to a pathway of allosteric regulation in this non-redundant transporter central to *S. aureus* virulence.

## Results

### Production and in vitro characterisation of *S. aureus* TarGH

Full-length TarG (residues 1–270 with an N-terminal polyhistidine tag) and TarH (residues 1–264) (Supplementary Fig. 1) from the methicillin-resistant *S. aureus* clinical strain USA300 were co-expressed in *Escherichia coli* or *Lactococcus lactis* (see "methods"). The resulting $TarG_2H_2$ heterotetramer was purified in the presence of LMNG and validated by gel electrophoresis and size exclusion chromatography coupled with multiangle static light scattering (Supplementary Fig. 2). Complexes from either the gram-negative or gram-positive expression systems behaved similarly in terms of solubilisation, oligomerization and activity, showing the same retention volume in size exclusion chromatography and consistent ATPase activity ($V_{max}$ ~ 70 mol Pi/min mol protein; Fig. 1c), mirroring prior reports for *E. coli* expressed *S. aureus*[24] or *A. herbarius*[30] TarGH.

To further probe the molecular details of its mechanism of action and inhibition, we determined cryo-EM structures of *S. aureus* TarGH bound to ATP analogue ATPγS alone or inhibitor targocil-II in the presence of either ATP analogues AMP-PNP or ATPγS (Supplementary Figs. 3–6, Supplementary Table 1).

### TarGH is a type V ABC transporter with an internal electropositive cavity compatible with WTA polymer binding

The cryo-EM structure of *S. aureus* TarGH in complex with ATPγS was determined to 2.3 Å resolution (Fig. 1d, Supplementary Fig. 3). One of the highest resolution ABC transporters yet determined by cryo-EM, the quality of the reconstruction permitted confident modelling of the complete sequences of TarG and TarH.

TarGH is designated a type V ABC transporter, a recently adopted classification based on TMD folds[31]. Each TarG TMD is comprised of six TM helices, N-terminal interface helices (IF1/2), two extracellular re-entrant helices (EH1/2), a cytoplasmic coupling helix (CpH) and a short cytoplasmic C-terminal interface helix (IFc) (Fig. 1d, Supplementary Fig. 1). The two TarG protomers dimerise with an interface area of ~1200 Å$^2$ (PISA[32]), forming an extensive internal cavity with dimensions and conserved electropositive nature complimentary to the soluble portion of its anionic polymer substrate (Fig. 1d). The

extracellular facing roof of the cavity is lined by the Arg186/Arg186' symmetric pair and sealed by a constriction immediately above formed by conserved aromatic residues on TM1/5 of both TMDs (Tyr51, Phe55, Phe189, Tyr190) (Figs. 1d, 2b).

The TarH NBDs (dimer interface area of ~1400 Å$^2$) each consist of a RecA-type ATP-binding core and ABC transporter specific domains ABCα and ABCβ. These harbour the characteristic functional motifs involved in ATP binding and hydrolysis including Walker A (Gly57-Ser64) and Walker B (Ile164-Asp168), D-loop (Glu169-Asp175), H-switch (His201), ABCβ A-loop (Tyr14) and the ABCα signature motif ($_{144}$YSSGM$_{148}$; Supplementary Fig. 1), highly conserved in TarH and other lipid flippases and notably deviating from the canonical LSGGQ[33,34]. We observe an additional lipid flippase specific gating motif (GM) comprised of α1/α11 (α1 was termed the gating helix (GH) in related ABC transporter WzmWzt[35], see below) and β2/3/12. The GH interacts with the loop connecting IF2 and TM1 (residues 31–35, denoted as the LG-loop[35]) of the opposing TarG protomer (Fig. 2a, c) and we propose plays a role in binding of the lipid-linked substrate (see below).

Analysis of structural similarity with DALI[36] indicates the closest match is WzmWzt from gram-negative *Aquifex aeolicus* (~41% sequence identity between TMDs TarG and Wzm, Z score 25.5; Supplementary Fig. 1). WzmWzt also transports a $C_{55}$-PP-linked substrate for synthesis of O antigens in the periplasm[37] although the glycan polymer is largely neutral and chemically distinct form the anionic substrate of TarGH. Structures of WzmWzt have been determined in various functional states[35,38–40]. However, in the absence of a lipid-linked substrate bound structure, the molecular details of the substrate loading state and flipping mechanism remain unclear.

### Ordered LMNG glycolipids suggest potential binding sites of the WTA substrate at both cytoplasmic and extracellular faces of TarGH

In this ATPγS-bound state, three well-ordered LMNG molecules are observed (Fig. 2a–c). Two are bound in symmetrically disposed wedge-shaped clefts formed at the inner leaflet/cytosol interface between the TarH GH, TarG IFc and the LG-loop on the opposing TarG protomer (denoted LMNG$_C$; Figs. 1d, 2a, c). The two acyl tails of LMNG$_C$ extend along the TarG-TarG interface while one of the maltose moieties is partially inserted into an electropositive lateral channel between TarH and TarG IFc that leads to the central cavity, and the other projects out to solvent (Fig. 2c). The third ordered LMNG is observed on the extracellular face of TarG, on the dimer rotation axis (LMNG$_E$; Fig. 1d, Fig. 2a, b). The acyl chains run down opposite sides of the TarG interface between TM1 and the TM5'-EH1' linker (denoted as the EL-loop) while the maltose groups extend toward solvent. The amphipathic, disaccharide nature of LMNG is similar to the $C_{55}$-PP-GlcNAc-ManNAc of the lipid-linked WTA substrate, albeit importantly lacking the negatively charged pyrophosphate group (Supplementary Fig. 7; see "discussion"). We propose therefore that the observed LMNG binding locations on the inner and outer leaflets are potentially reflective of the cytosolic entrance and extracellular exit points of the pre and post flipped substate.

In addition to the ordered LMNGs as above, there are multiple further densities surrounding the TMDs potentially consistent with ordered lipids, or parts thereof. The most defined of these extends along a conserved hydrophobic cleft formed by TM5, TM6 and the reentrant helices EH1/2 (Supplementary Fig. 7).

### The TarH D-loop adopts a catalytically incompetent conformation when bound to ATPγS

The level of detail at 2.3 Å resolution allowed unambiguous modelling of key active site features that show it to be in a non-optimal configuration for ATP hydrolysis. ATPγS is well resolved at the TarH dimer interface, forming characteristic interactions with the ABC motifs described above (Fig. 2e). Notably, the catalytically inert γ-(thio)

phosphate forms a hydrogen bond (2.7 Å) and electrostatic interaction with His201, proposed to play an essential mechanistic role in ATP hydrolysis in other ABC transporters[41]. Density for the catalytic

magnesium is observed, coordinated with typical octahedral geometry by the γ-(thio)phosphate, Ser64 side chain hydroxyl and three waters. The Walker B Asp168, implicated in metal coordination in many ABC

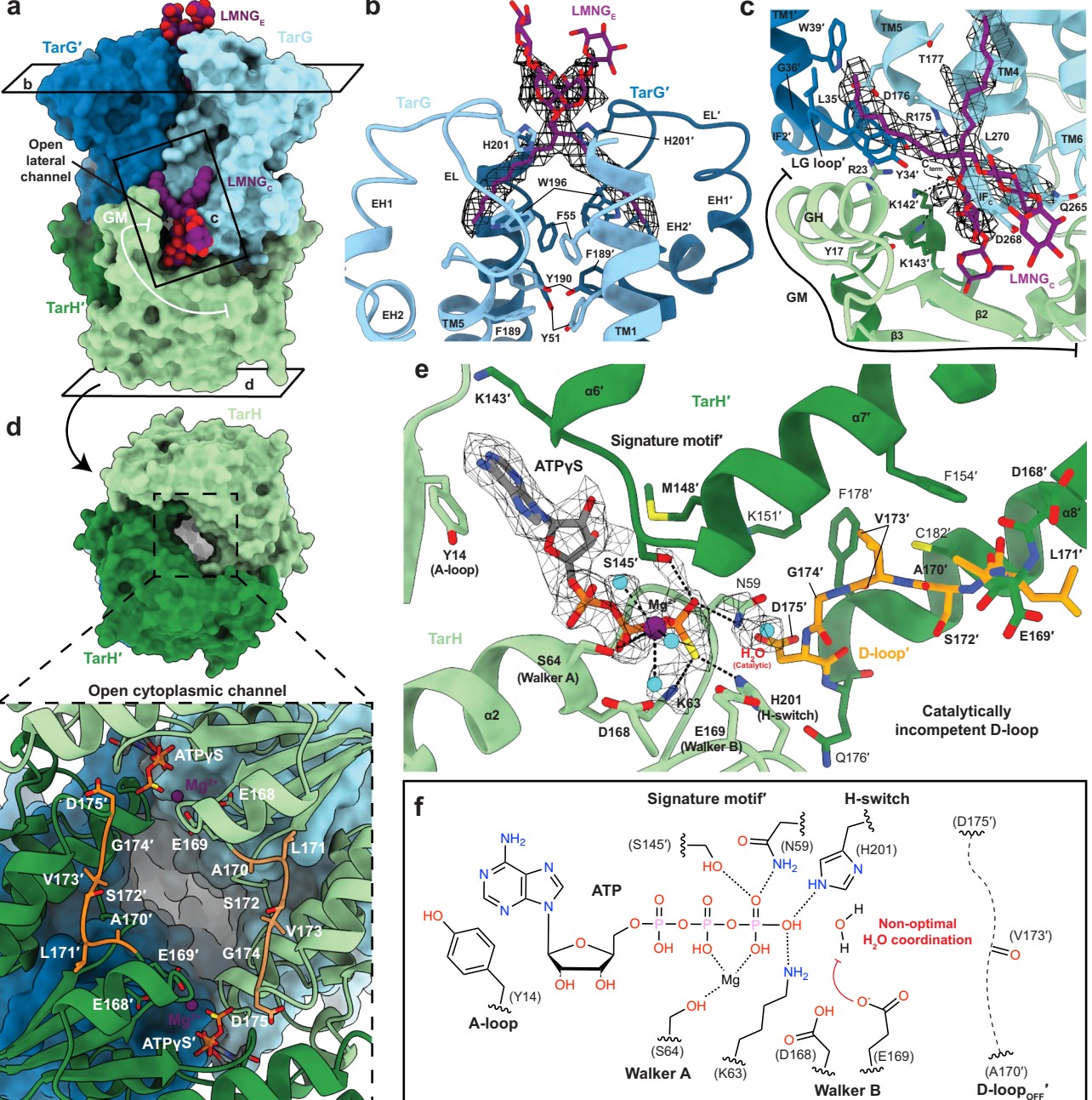

**Fig. 2 | Structural features of _S. aureus_ TarGH in a pre-catalytic state with ATPγS and ordered LMNG at 2.3 Å resolution. a** Structure of ATPγS-bound _S. aureus_ TarGH, surface representation coloured with LMNG shown in purple as spheres with heteroatom colouring. Boxed regions denote important structural features with expanded views in the corresponding lettered panels. **b** Cryo-EM density of extracellular bound LMNG_E (purple sticks, heteroatom colouring; map-threshold: 0.152) bound at the extracellular dimerisation interface of TarG (light/dark blue ribbon). The extracellular LMNG binding pocket straddles the inter-TarG aromatic constriction (Tyr51, Phe55, Phe189, Tyr190) with interface residues shown as sticks with heteroatom colouring. **c** View of the cytoplasmic LMNG_C in the gate pocket at the interface of TarG and TarH. Coloured as in (**b**) with interacting and functionally important residues shown as sticks with heteroatom colouring and LMNG cryo-EM density depicted as grey mesh (map-threshold: 0.177). **d** Surface view from base of TarGH illustrating the channel extending through the TarH dimer from the cytoplasm into the intramembrane cavity within TarG, coloured as in (**a**). Boxed region

provides a zoomed-in perspective of the D-loop_OFF conformation that defines the channel (shown as gold cartoon/sticks with heteroatom colouring). **e** View of the TarH active site with bound ATPγS. Coloured as in (**d**) with ATPγS shown as sticks with heteroatom colouring (carbons grey). Density for ATPγS depicted as a grey mesh (map-threshold: 0.17). Catalytic magnesium and coordinating waters are shown as purple and cyan coloured spheres, respectively. Canonical ABC transporter motifs (A/D- loops, H-switch, signature motif) are labelled and important residues shown as sticks. The D-loop of the opposing chain is shown as sticks and coloured light orange. The catalytic water, shown as a cyan-coloured sphere, is non-optimally positioned, displaced from the line-reaction plane of the γ-phosphate and catalytic Glu169, likely due to the loss of the coordinating D-loop carbonyl from Val173'. **f** Schematic representation of the TarH active site illustrating the D-loop_OFF conformation leads to a catalytically incompetent active site structure with the backbone carbonyl of Val173' oriented away from the active site and unable to coordinate the catalytic water.

transporters[42], is too distant to directly interact with the magnesium. Interestingly, the neighbouring Glu169, the typical general base for activation of the water in ATP hydrolysis[43,44], lacks sidechain density despite the high resolution. This could suggest it is either not well ordered or that it is in a negatively charged carboxylate state invisible in the Coulomb potential map (a phenomenon described by Yeager and colleagues[45]), consistent here with its general base role.

The highly conserved Asp175 of the D-loop ($_{170}$ALSVG**D**$_{175}$) maintains the typical inter chain hydrogen bond with the backbone amide of Asn59′ connecting the Walker B motif of one active site to the Walker A of the other. However, the preceding D-loop residues deviate from the near helical conformation commonly observed in other ABC transporters[46–48] and are, instead, more closely associated with the downstream helix α8 (the D-helix) and ABCα domain. The aliphatic side chain of Val173 is inserted into a hydrophobic pocket, which includes invariant aromatics Phe154 and Phe178 (Fig. 2e), with its backbone carbonyl flipped away from the active site and no longer in position to coordinate the catalytic water during ATP hydrolysis (we designate this D-loop$_{OFF}$; Fig. 2f)[46]. Reflecting this, a water is observed in the approximate region, but the supporting density is extended and in a non-optimal position due to the D-loop$_{OFF}$ conformation (Fig. 2e). An additional consequence of this rearranged D-loop$_{OFF}$ is the creation of a solvent exposed cytoplasmic channel of ~10 Å across at its narrowest that runs through the NBD dimer, continuous with the electropositive intramembrane cavity between the two TarG protomers (Fig. 2d).

## Inhibitor targocil-II binds to the extracellular site of TarG

Targocil-II was developed using a high throughput screen and SAR optimisation[24] (Fig. 3a). In our hands, it inhibited ATPase activity with an IC$_{50}$ value of 6.5 ± 1.2 μM (Fig. 3b). An ATP analogue was required for measurement of direct binding using microscale thermophoresis with AMP-PNP leading to approximately an order of magnitude tighter binding of targocil-II compared to ATPγS (Fig. 3c). Based on this, we determined the cryo-EM structure of TarGH in complex with targocil-II and AMP-PNP at 2.9 Å resolution (Fig. 3d, Supplementary Fig. 4).

Clearly resolved densities for two symmetrically disposed targocil-II molecules are observed at the extracellular interface of the TarG dimer (Fig. 3d, e). Each is bound in a pocket formed by the C-terminal residues of TarG TM1 and TM5, the N-terminus of EH1 and the intervening EL-loop, as well as TM1′ of the opposing TarG promoter (Fig. 3e). The planar polycyclic furanocoumarin core is sandwiched between conserved Trp196 on the EL-loop and the aliphatic side chains of Val54 and Ile59 on TM1′. The side chain of Phe191 on TM5 further stabilises binding, with the main chain torsion angles and Cα position of the largely conserved Gly56 on TM1′ completing the close packing around the furanocoumarin core. The chlorophenyl group extends toward the outer surface of the TarG interface, interacting with the side chains of Ile207 on EH1 and Leu57 on TM1′. Notably, the negatively charged L-proline carboxylate on the opposite end of targocil-II interacts with the electropositive side chains of conserved Arg60 on TM1 and Lys199 on the EL-loop (Fig. 3e, Supplementary Fig. 1). The position of this group was important in the development of targocil-II, with the L- amino acid providing an order of magnitude increase in potency compared to the D-stereoisomer and proline a twofold improvement over alanine[24]. This interaction could also explain the eightfold lower activity of targocil, which has no functional counterpart, compared to targocil-II (MICs for *S. aureus* are 1 and 0.125 μg/mL, respectively[24]).

The targocil-II bound structure allows us to analyse existing resistance mutants and species-specific variation of inhibitor efficacy. The localisation of prior in vitro resistance mutants are supportive of our experimentally observed targocil-II binding site[24]. These include Val54 (V54L), Phe55 (F55L), Leu195 (L195F) and Trp196 (W196C/L), which all form hydrophobic interactions with the inhibitor (Fig. 3e, f). A similar set of resistance mutants were generated for other classes of inhibitors including targocil[23,27] suggesting a related binding

mechanism. We note that in the prior *A. herbarius* TarGH study[30], resistant residues were used to define a hydrophobic pocket to guide computational docking of targocil after removal of the EL-loop. However, two of the mutants—Trp73 and Phe82—do not directly interact with targocil-II here while EL-loop residues Trp196 and Lys199 form key interactions as above (Fig. 3e, f).

Targocil-II was reported to lack action against the more sequence diverse *Bacillus subtilus*[24], which may be fortuitous given its common role as a gut commensal, but activity against other species has not been reported. To explore this further, we performed growth assays using a panel of *S. aureus*, *B. subtilis*, *S. epidermidis* and *S. pneumoniae* strains, the latter two species of particular clinical importance (Fig. 3g). Targocil-II showed potent inhibition of all *S. aureus* strains tested (minimum inhibitory concentration (MIC) < 0.5 μg/mL; previously reported MIC = 0.125 μg/mL[24]) but was not effective against *B. subtilis* as prior, nor the *S. epidermidis* or *S. pneumoniae* tested here (MICs > 64 μg/mL; Fig. 3g).

The targocil-II binding site, including the positioning of positively charged residues for electrostatic interaction with the targocil-II prolyl carboxylate, is well conserved (Supplementary Fig. 1). The only observed differences structurally reside on the EL-loop: specifically, Phe191 (interacts with the targocil-II furan ring; methionine in *S. epidermidis*) and Ile203 (interacts with chlorophenyl; valine in *S. epidermidis*) (Fig. 3h). This seemingly subtle loss of optimal van der Waals and hydrophobic interactions in this binding region appears to result in sufficient loss of binding to diminish targocil-II efficacy (Fig. 3g, h). These observations give hope that targocil-II variants can be modified to accommodate for these small and localised differences to target additional clinically important pathogens.

## Targocil-II binding causes local and global conformational changes in TarGH

Targocil-II binding overlaps with the LMNG$_E$ observed in the prior ATPγS-bound structure, with the relatively nonpolar furanocoumarin bound in the same hydrophobic pocket as one of the acyl tails of LMNG. Compared to LMNG$_E$, however, targocil-II binding induces extensive local conformational changes around the EL-loop (Fig. 4b, c). The Trp196 side chain is flipped out of the pocket to accommodate the deeper penetration of the coumarin core and Phe191 also adopts a different rotamer, projecting toward TM1′ to interact with the furan ring. Finally, the targocil-II L-proline group displaces His201 (Cα shift ~12 Å) with the Lys199 side chain flipping ~180° to electrostatically engage the carboxylate. Together, these changes result in a large reorientation of the EL-loop (Pro198-Ser204) and a subtle shift of EH1, EH2, TM6 and IF1 (Fig. 4b, c).

In addition, more remote changes are observed in TarH compared to the ATPγS-bound state without inhibitor with notably the TarH D-loop now adopting the canonical near helical turn (D-loop$_{ON}$) that is conserved with other ABC transporters[46–48]. To investigate if these differences were a result of targocil-II binding or the choice of ATP analogue, we determined structures of TarGH produced in *L. lactis* in complex with ATPγS alone and with targocil-II (Supplementary Fig. 5, 6, 8). The structure in complex with ATPγS alone (2.9 Å resolution) is near identical to the *E. coli* produced protein (RMSD 0.58 Å and 0.52 Å across TarH and TarG Cα-atoms, respectively; Supplementary Fig. 8). In the presence of ATPγS and targocil-II, which is bound near identically to the AMP-PNP structure as above, we observe two approximately equally populated states of D-loop$_{ON}$ (3.0 Å resolution) and D-loop$_{OFF}$ (2.7 Å resolution) (Supplementary Figs. 6, 8). This supports that the shift to the TarH D-loop$_{ON}$ state is a result of targocil-II binding TarG and not the difference in ATP analogue. However, the mixed population also suggests that D-loop$_{ON}$ is less energetically favourable in the presence of ATPγS compared to AMP-PNP, where we only observe D-loop$_{ON}$, possibly due to steric or charge differences of the thiophosphate that could repel the approach of Val173. This

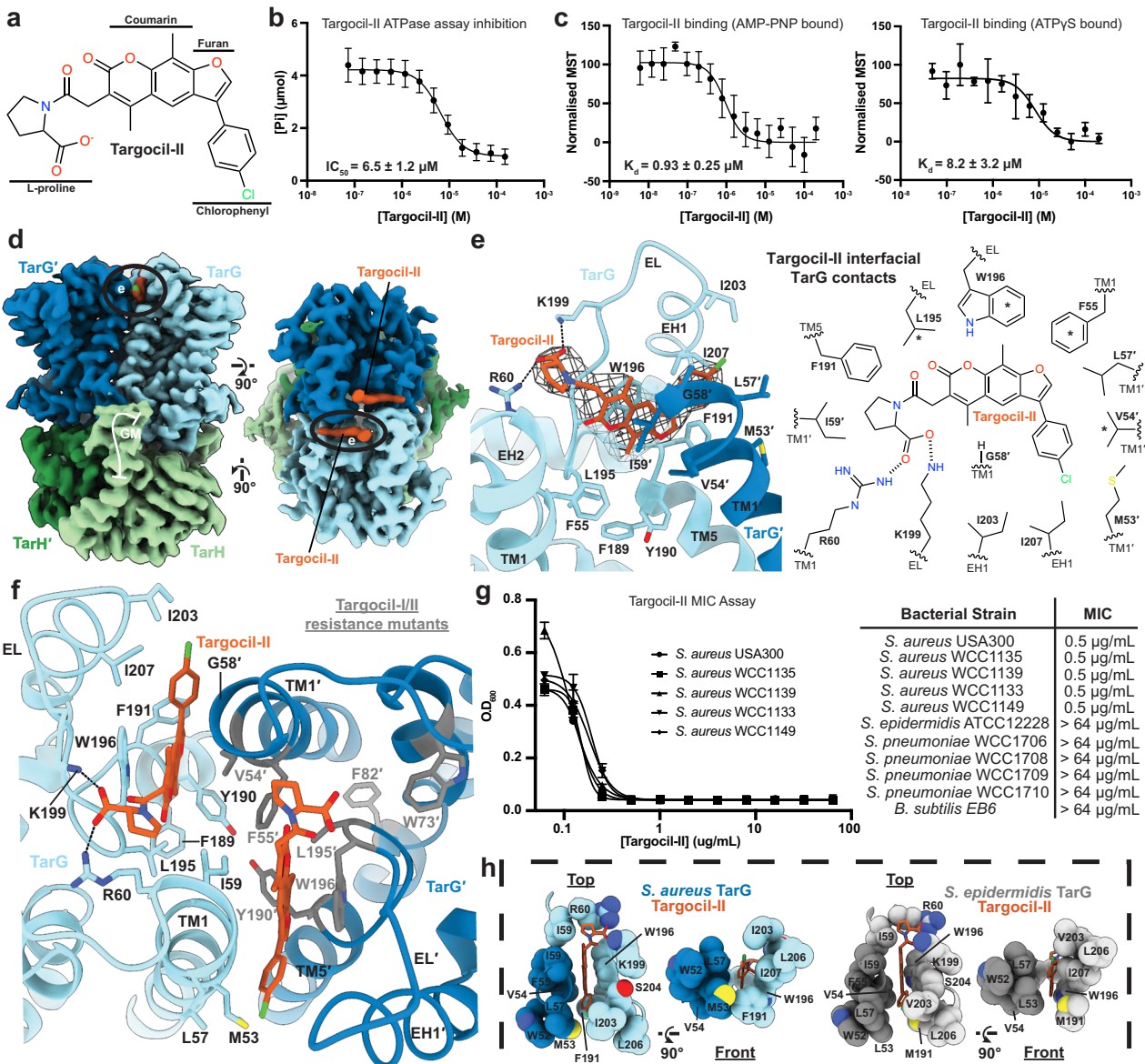

**Fig. 3 | Structural, kinetic and cellular analysis of targocil-II inhibition of *S. aureus* TarGH.** **a** Chemical structure of targocil-II. **b** Inhibition of TarGH ATPase activity by targocil-II, as measured by a malachite green assay. Data points represent the mean from three independent trials ± standard deviation. Half-maximal inhibitory concentration (IC$_{50}$) is reported as mean ± 95% confidence interval. **c** Microscale thermophoresis binding of AMP-PNP (left) or ATPγS (right) bound *S. aureus* TarGH to targocil-II. Experimental data is reported as the mean from at least three independent trials ± standard deviation and dissociation constant (K$_d$) reported as mean ± 95% confidence interval. **d** Cryo-EM reconstruction of targocil-II and AMP-PNP bound *S. aureus* TarGH (2.9 Å resolution). TarG is coloured in dark/light blue, TarH in dark/light green and targocil-II in dark orange. Targocil-II is circled. **e** View of the targocil-II binding site on TarG on left. Coloured as in (**d**). The cryo-EM density (map-threshold: 0.06) for targocil-II is depicted by a dark grey mesh. 2D chemical representation of binding site shown on right. Non-carbon atoms are coloured distinctly, and side chains are labelled with corresponding TarG structural feature. An asterisk (*) is used to denote positions of targocil-II resistance mutants. **f** Two-fold symmetric targocil-II binding sites at the extracellular portion of the TarG dimerisation interface with identified targocil and targocil-II resistance mutants coloured in grey. Targocil-II and interacting residues are shown as sticks with heteroatom colouring. TarG structural elements labelled. **g** Bacterial growth assay for gram-positive *S. aureus*, *S. epidermidis*, *S. pneumoniae* and *B. subtilis* strains in presence of targocil-II. Left: growth curve for inhibited strains. Cellular assay data from triplicate measurements is reported as the mean ± standard deviation. Right: table of minimum inhibitory concentrations (MIC). **h** Targocil-II binding site for *S. aureus* (left; experimental) and *S. epidermidis* (right; AlphaFold model). Residues defining the binding site shown as spheres with differences at position 53 (Met vs Leu), 191 (Phe vs Met) and 203 (Ile vs Val) around the chlorophenyl binding pocket.

hypothesis is also supported by the observed weaker binding of targocil-II in the presence of ATPγS (Fig. 3c).

Comparing the D-loop$_{OFF}$ and D-loop$_{ON}$ states, global changes are most evident in TarH, with a subtle movement away from TarG and ~15° clockwise rotation around its symmetry axis as viewed toward the membrane impacting the TarG-TarH interface (~790 Å² vs ~1010 Å² in absence of targocil-II; Fig. 4a, b). The GH is also rotated toward TarG, collapsing the wedge-shaped gating pocket where LMNG$_C$ was bound

in the absence of targocil-II (Fig. 2c, Supplementary Fig. 7), with no LMNG now observed in its presence (Fig. 4a, Supplementary Fig. 7). Within TarH there is a significant reorientation of two loops and associated closure of the central channel between protomers (Fig. 4a, b, d). The first is the loop connecting strand β7 downstream of the Walker A motif and helix α3 at the start of the ABCα domain, which is in close proximity to the coupling helix (CpH) of TarG (Fig. 4d). This loop has historically been referred to as the Q-loop based on a conserved

glutamine[49,50] and proposed to have a role in the coupling of TMD and NBDs[51–53]. The glutamine is not conserved in TarH, which instead has a conserved isoleucine at this position (Ile92; Supplementary Fig. 1).

Within the loop, residues Ile90-Gly98 undergo the biggest shift with Ile92 moving ~6 Å (Cα distance) toward the Walker B motif and catalytic Glu169 (Fig. 4d). The second is the D-loop (Glu169-Asp175), which

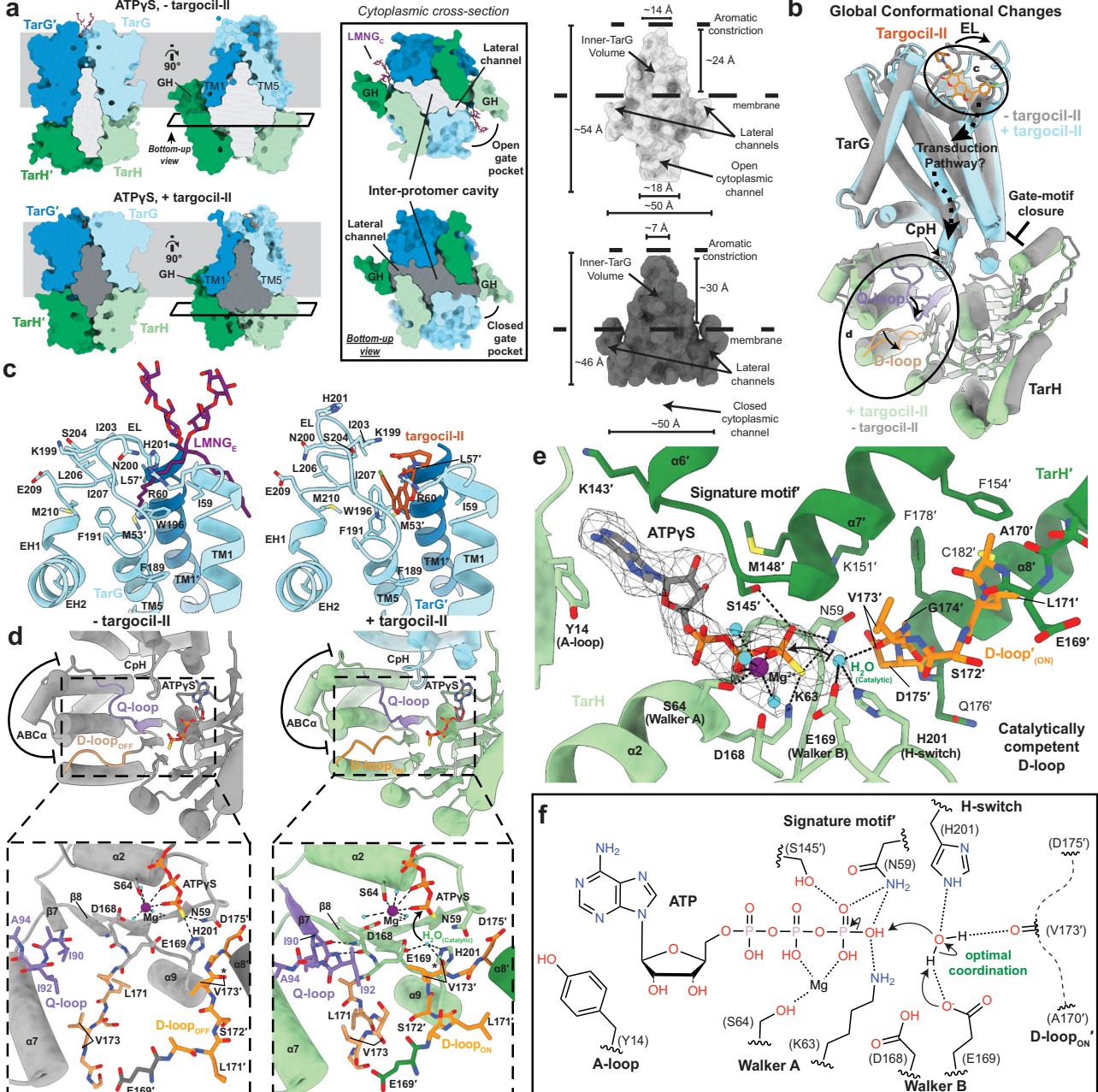

**Fig. 4 | Targocil-II induced conformational changes and allosteric ATPase activation. a** Internal cavity analysis for *S. aureus* TarGH in the absence (top, light grey) or presence (bottom, dark grey) of targocil-II. Analysis was performed with 3 V software using a 9 Å large probe and 2 Å small probe size. TarG and TarH are coloured light/dark blue and green, respectively, with bound LMNG and targocil-II shown as sticks with heteroatom colouring (purple and orange, respectively). Boxed regions represent lateral cross sections viewed from cytoplasm. Internal cavity volume, dimensions and features are labelled on the right. **b** Cartoon overview of conformational changes induced by targocil-II binding. Structure in absence and presence of targocil-II shown in grey and blue/green, respectively. Solid arrows indicate observed conformational changes with dashed arrows representing the hypothesised direction of allosteric communication. Circled regions represent significant conformational changes shown in (**c**, **d**). **c** Structure of the extracellular binding pocket in absence (left; with bound LMNG_E) and presence (right) of targocil-II. Targocil-II binding induces conformational changes in the EL-loop connecting TM5 and EH1. TarG and bound ligands coloured as in (**a**).

**d** Allosteric coupling of targocil-II binding with canonical Q/D-loop movement in the TarH ATPase domain bound to ATPγS. TarGH in the absence (left, grey) or presence (right, blue/green) of targocil-II with the Q-loop shown in light purple, the D-loop in orange and ATPγS as grey sticks with heteroatom colouring. Boxed out regions show variation in Q-/D- loop interactions, with both loops shown as sticks with colouring as above. Dashed lines represent interactions with magnesium coloured purple, coordinating waters coloured cyan, and catalytic water coloured cyan and labelled. Asterisk denotes position of the D-loop Val173 carbonyl which reorients into the active site in the presence of targocil-II. **e** TarH active site with bound ATPγS. In the presence of targocil-II bound to TarG, this adopts a catalytically competent active site structure. Cryo-EM density for ATPγS, magnesium and coordinating waters (map-threshold: 0.35) are shown as a dark grey mesh.
**f** Schematic representation of the TarH active site illustrating the D-loop_ON conformation leads to a catalytically competent structure with the backbone carbonyl of Val173 oriented toward the active site and able to coordinate the catalytic water.

now adopts the canonical near helical turn that is conserved with other ABC transporters[46–48]. This puts the active site in a catalytically competent conformation (D-loop$_{ON}$; Fig. 4e, f) with the backbone carbonyl of Val173 flipped and ideally positioned to coordinate and activate a water for nucleophilic attack together with Glu169. We note that density for the side chain of Glu169, not observed in the 2.3 Å ATPγS structure, is now clearly resolved (Supplementary Fig. 7). Q-loop residues Ile90 and Ile92 in the D-loop$_{ON}$ state now form backbone β-strand hydrogen bonds with Glu169 that could stabilise its position. These changes appear concerted with Q-loop Ile92 contributing to the hydrophobic pocket that stabilises D-loop Val173 in the D-loop$_{OFF}$ conformation (Fig. 4d–f). In the D-loop$_{ON}$ state, a water is observed coordinated by the Glu169 side chain carboxylate and Val163 main chain carbonyl consistent with a role in catalysis, with a ~3.9 Å approach distance to the γ-phosphorous ideal for an in-line reaction coordinate[54] (Fig. 4e, f).

## Discussion

In *S. aureus*, the ABC transporter TarGH plays an essential role in transport of the lipid-linked WTA anionic polyribitol polymer, $C_{55}$-PP-GlcNAc-ManNAc-GroP-(RboP)$_{n<40}$, across the membrane for subsequent glycosyltransfer of the soluble polymer onto PG strands of the bacterial cell wall (Fig. 1a, b). Here, we have determined structures of *S. aureus* TarGH bound to two ATP analogues in distinct pre-hydrolytic[55] states. The first, in complex with ATPγS, shows the TarH D-loop in a conformation (D-loop$_{OFF}$) that appears non-competent for ATP hydrolysis. The second, bound to either AMP-PNP or ATPγS, and inhibitor targocil-II, which binds to the extracellular dimerisation interface of TarG, adopts the canonical TarH D-loop conformation (D-loop$_{ON}$) typically observed in other ABC transporters that is well positioned for hydrolysis. In addition to these mechanistic snapshots, the bound targocil-II provides the atomic details of how it acts as a substrate mimic to disable the necessary and non-redundant TarGH translocation of WTA polymers and in turn *S. aureus* virulence.

The D-loop$_{OFF}$ and D-loop$_{ON}$ structures suggest the TarH active site is capable of significant conformational changes throughout the ATPase catalytic cycle. Although not previously described in TarGH or other ABC transporters, D-loop$_{OFF}$ is independently supported in both the lower resolution structure of the *A. herbarius* TarGH catalytic E169Q mutant[30] and in a cryo-EM analysis of WzmWzt undergoing ATP turnover when vitrfied[40]. For *A. herbarius* TarGH (EMD-9790), the open channel through the TarH dimer characteristic of D-loop$_{OFF}$ is clearly observed in the map even though the resolution prevented accurate atomic modelling of the D-loop (Supplementary Fig. 9). An endogenous nucleotide, likely ATP, was overlooked and the structure is consistent with the ATPγS-bound D-loop$_{OFF}$ state here. The structure was also determined using detergent DDM with nothing bound in the gate pocket and the GH much more poorly resolved, suggesting the D-loop$_{OFF}$ state is not specifically induced by LMNG. In WzmWzt, inspection of the map from a sample hydrolysing ATP (EMD-27564), which is an average of all transporter states in the data used to reconstruct it, shows evidence for both D-loop$_{ON}$, as modelled and D-loop$_{OFF}$ similar to our TarH structure (Supplementary Fig. 9). The conformational transition between D-loop$_{OFF}$ and D-loop$_{ON}$ involves a peptide-flip of conserved Val173-Gly174 that places the backbone carbonyl of Val173 in an optimal position to coordinate the catalytic water with Glu169 (Figs. 2e, f, 4e, f). Similar peptide-flipping involving X-Gly or Gly-X motifs has been proposed as a means of allosteric enzyme activation, where larger scale conformational changes are coupled to local backbone reorientations[56]. In addition to the D-loop, we observe a concerted movement of the Q-loop at the interface of the TMDs and NBDs. The Q-loop glutamine is instead a conserved isoleucine in TarH (Ile92; Supplementary Fig. 1), which appears important in both D-loop$_{OFF}$ and D-loop$_{ON}$ states. In D-loop$_{OFF}$, Ile92 forms part of the hydrophobic pocket that sequesters D-loop Val173 and in D-loop$_{ON}$, it

engages the active site and, along with Ile90, forms stabilising backbone β-strand interactions with the Walker-B motif and catalytic Glu169 (Fig. 4d). Both the D- and Q- loops have been implicated in the allosteric communication between TMDs and NBDs. A molecular dynamics simulation in ABC transporter Sav1866 predicted a retraction and approach of the D-loop to the *trans* ATP γ-phosphate, specifically the carbonyl of the equivalent residue to Val173, and also a major role of the Q-loop glutamine in defining a catalytically capable active site[57,58]. This has been supported by structural studies of the D-loop[59–61] and Q-loop[62]. However, to our knowledge, such a drastic and concerted conformational change as observed for TarGH has not previously been documented. The Q-loop (Ile90 and Ile92) and D-loop (Val173 and Gly174) features are also conserved in WzmWzt (Ile are replaced with Leu; Supplementary Fig. 1) suggesting a common physiological role. We suggest that the D-loop$_{OFF}$ state in the absence of substrate could provide a mechanism to minimise spurious waste of ATP when not needed for WTA polymer flipping.

The bound LMNG and targocil-II suggest binding sites for the lipid-linked polymer WTA substrate, which share in varying degrees similar physicochemical properties (Supplementary Fig. 7). In the ATPγS-bound D-loop$_{OFF}$ structure, an LMNG (LMNG$_C$) is bound in the pocket between the TarH GH and TarG IF$_C$ (Fig. 2c). Based on its location, size and shape, the equivalent pocket in WzmWzt was also hypothesised to be a substrate binding site (although not directly captured for characterisation)[35]. In the targocil-II/AMP-PNP-bound D-loop$_{ON}$ structure, movement of TarH and the GH has collapsed this pocket and no LMNG is observed. We speculate therefore that the LMNG$_C$-bound (D-loop$_{OFF}$) structure represents a substrate loaded state with the cytoplasmically oriented lipid-linked polymer captured at the inner leaflet of the membrane but prior to polymer insertion and translocation. Targocil-II or LMNG (LMNG$_E$) bind to a pocket at the extracellular surface of the TarG dimerisation interface (Figs. 2b, 3e, f) with targocil-II, but not LMNG, inducing significant local rearrangements. We speculate that targocil-II is mimicking binding of the $C_{55}$-PP carrier, with the furanocoumarin core and chlorophenyl group mimicking the isoprenyl repeats and the L-proline carboxylate mimicking the negative charge, with no counterpart in LMNG to facilitate the same electrostatic interactions (Supplementary Fig. 7). We propose this is the post-translocated substrate binding site and, in absence of an opening into the intramembrane cavity, reflects a state where the polymer has completed translocation, but the lipid carrier is still associated with TarG.

Thus, these conformational changes upon targocil-II binding provide evidence to support how substrate binding to the extracellular region of the TMDs may be allosterically coordinated with ATPase activity in the cytoplasmic NBDs via changes in the TarH Q- and D-loops. Based on the catalytically competent D-loop$_{ON}$ active site structure in the presence of targocil-II, we speculate that a post-translocated substrate could modulate ATPase activity cycle to, for example, induce substrate dissociation or reset to a state primed for substrate capture.

The mechanism of substrate translocation from the cytoplasm/inner leaflet to extracellular/outer leaflet remains unclear. The electropositive interior supports the anionic WTA polymer crossing the hydrophobic membrane through the central cavity with mutations in TarG residues lining the cavity previously shown to impact WTA levels[30]. Structural snapshots of WzmWzt with separated NBDs[40] or an open aromatic constriction[40] (Supplemental Fig. 9) hint at how the polymer might access and exit this cavity. However, how the lipid anchor reorients is less obvious with its hydrophobic nature less compatible with the amphiphilic cavity interior. We observe multiple extended lipid densities surrounding the TarG TMDs (Supplementary Fig. 7). The most defined of these is in a hydrophobic pocket below the TarG renentrant helices EH1/2 (Supplementary Fig. 7). The reentry helix motif is an exporter specific regulatory element shown in similar ABC

exporters to assist in lipid flipping[63–65]. This motif has also been implicated in the specific recognition of prenyl lipids[66]. A specific binding site for the undecaprenyl ($C_{55}$) carrier lipid on the surface of TarG could facilitate an increase in local concentration of the polymeric substrate prior to translocation. The relative disposition between this lipid density and the bound $LMNG_C/LMNG_E$/targocil-II would permit access to both proposed cytoplasmic/extracellular substrate binding sites from a fixed $C_{55}$ anchor position. In this scenario, where the $C_{55}$ lipid carrier was sequestered on the surface of TarG, the linkage unit would reorient through the TarG dimerisation interface as also proposed for WzmWzt[38].

TarGH is essential, non-redundant and extracellularly accessible, collectively representing an attractive target for antimicrobial development with inhibitors shown to be bacterial static and working synergistically with β-lactams[26–29]. Our structures of TarGH in complex with targocil-II reveal its mechanism of binding and inhibition and provides a template for further structure-based drug design. The relative conservation of the targocil-II binding site amongst other gram-positive pathogens, including *Staphylococcus epidermidis*, *Enterococcus faecalis* and *Listeria monocytogenes*, add to its potential promise for development as a new class of anti-infective drugs for antibiotic resistant infections. One question raised here is how targocil-II inhibits TarGH if at the same time inducing an active conformation of the TarH active site D-loop as we observe. TarGH uses the energy derived from ATP hydrolysis to drive substrate transport across the membrane. Assuming overlapping binding sites as we propose, targocil-II could sterically block substrate access to the extracellular binding site. We also observe a reduction of ATP hydrolysis in absence of translocation substrate, as measured by release of inorganic phosphate, albeit not complete (Fig. 3c). We envisage this allosteric inhibition could arise from interruption of the conformational changes required to fulfil the dynamic ATPase hydrolysis cycle. In this case, hydrolysis could still occur, but the structural changes required for release of the reaction products or reset to the ATP binding state for subsequent cycles could be inhibited with the same result of reduced ATP turnover and ultimately inhibition. We also note that the targocil-II bound TarGH heterotetramer is locked in a symmetric state. For the WTA lipopolymer substrate, however, it is hard to imagine that both extracellular sites could be similarly occupied, with the dimensions of the inner aromatic constriction likely allowing passage of only one substrate at a time. This would fit in with an asymmetric conformational cycle as has been proposed for other ABC transporters[67] such that only one TarH protomer is activated at a time through interaction of the WTA substrate with the EL-loop and downstream conformational effects. Targocil-II may break this asymmetry and lock the transporter in the activated ATP bound form but unable to transition to the next structural state in the cycle of hydrolysis and reset.

## Methods

### Plasmid Construction and expression in *Escherichia coli*

USA300 *S. aureus* TarG (residues 1–277; Uniprot: A0A0H2XIF1) and TarH (residues 1–264; Uniprot: Q2FJ01) were codon optimised for *E. coli* expression by TWIST Bioscience and cloned (Construct 1) into a modified pETDuet vector with a TarG N-terminal octa-histidine tag using restriction-free cloning[68] (Supplementary Table 2). Cloning was performed in electrocompetent DH10β *E. coli* grown in 2 × YT media supplemented with 34 μg/ml chloramphenicol ($Cm_{34}$) and verified by GENEWIZ (Azenta Life Sciences) sequencing. After verification of the DNA sequence, codon optimised *S. aureus* TarGH was transformed in *E. coli* C41[69] using a standard PT7-lacO (T7) expression system. Single colonies selected from $Cm_{34}$ plates were grown as seed cultures in 2 × YT media overnight and a 1:50 (v/v) addition of seed culture was used to inoculate 1 L of ZYP-5052 autoinduction media[70]. Expanded cultures were grown for 5 h at 37 °C followed by overnight protein expression at 24 °C. Cells were harvested by centrifugation and frozen in liquid nitrogen before storage at −80 °C until required.

### Plasmid construction and expression in *Lactococcus lactis*

For tandem expression of *S. aureus* TarG and TarH in *Lactococcus Lactis* (*L. lactis*), genomic *S. aureus* TarG and TarH (USA300 strain) were amplified with primers from Integrated DNA Technologies and PCR products cloned into a modified bicistronic pNZ8048 plasmid using restriction-free cloning (Supplementary Table 2). Synthesis of the bicistronic pNZ8048 template plasmid (Construct 2) was accomplished by amplification of the pNZ8148 (pNZDual[71]) bicistronic multiple cloning site 1/2 and subsequent linearisation into a pNZ8048[72] plasmid. Sequences corresponding to native *S. aureus* USA300 TarG was initially cloned into an N-terminal, thrombin cleavable hexa-histidine tag containing pNZ8048 plasmid (Construct 3) before subsequent reamplification and insertion into MSC1 for generation of a pNZ8048 Dual construct mimicking the *E. coli* pETDuet construct above. *S. aureus* TarH was subsequently amplified and inserted into MSC2 for tandem expression of tagless TarH and N-terminal tagged TarG (Construct 4). Cloning was performed in *E. coli* MC1061 cells grown in 2 × YT medium supplemented with 10 μg/mL chloramphenicol ($Cm_{10}$) before transformation of verified plasmids into electrocompetent *L. Lactis* pNZ9000 cells following standard protocols[73].

*S. aureus* TarGH was expressed in a *L. lactis* PNZ9000 cells through a nisin-controlled gene expression system following protocols optimised previously for the expression of *S. aureus* BlaR1[74,75]. Namely, single PNZ9000 colonies containing *S. aureus* TarGH were grown overnight in 250 mL of sterilised M17 medium (Difco) supplemented with $Cm_{10}$ and 0.5% w/v glucose at 30 °C. The overnight *L. lactis* culture was added to a 6L-fermenter (Applicon) under temperature control (30 °C) containing 5.5 L of autoclaved M17 media supplemented with 100 μM zinc sulphate, $Cm_{10}$ and 100 μL of antifoam 204 (SIGMA). During cell growth and expression, the medium was maintained at pH 7 by addition of ammonium hydroxide. *L. lactis* cells were grown to an optical density at 600 nm of around 1.5 before saturated nicin (1 mg/mL) was added to a final concentration of 5 ng/mL. Protein was expressed for 3 h post-nicin addition, collected by centrifugation and then flash frozen in liquid nitrogen before storage at −80 °C.

### TarGH purification

*S. aureus* TarGH produced from *E. coli* and *L. lactis* cultures was purified following an optimised protocol independent of expression source. Cell pellets were thawed, resuspended in lysis buffer (50 mM Tris pH 8, 500 mM NaCl, 10% glycerol) containing 10 μg/mL Lysozyme (Biobasic) and cOmplete, EDTA-free protease inhibitor cocktail (Sigma-Aldrich) and incubated for 45 min at 4 °C (*E. coli*) or 24 °C (*L. lactis*) with gentle stirring before DNase (25 μg/mL, Sigma-Aldrich) was added for an additional 15 min. Cells were subsequently passed thrice through a Constant Systems CF1 High Pressure Cell Disruptor (30k psi *E. coli*, 40k psi *L. lactis*) at 4 °C. Cell debris was pelleted by centrifugation at 12,000 × *g* for 30 min and the membrane containing supernatant was centrifuged for an additional hour at 180,000 × *g*. The isolated membrane pellet was homogenised in lysis buffer (50 mM Tris pH 8, 500 mM NaCl, 10% glycerol) using a glass Teflon Dounce homogeniser in aliquots derived from 2 L of culture.

*S. aureus* TarGH containing membranes were extracted with 1% Lauryl maltose neopentyl glycol (LMNG, Anatrace) at 4 °C before centrifugation at 180,000 × *g* for 30 min to pellet insolubilised membrane fractions. The supernatant was then supplemented with 15 mM imidazole pH 8.0 and incubated overnight with 2 mL Ni-NTA resin (Thermo Scientific) equilibrated in buffer A (50 mM Tris pH 8, 500 mM NaCl, 10% Glycerol, 0.03% LMNG). The incubated resin was settled and purified via gravity flow at roughly 0.5 mL/min. Removal of weakly associated contaminants was achieved by 20 mL washes of buffer A supplemented with 20, 40 and 60 mM imidazole and TarGH was

eluted with 20 mL of buffer A containing 250 mM imidazole. The protein was then concentrated to 3 mg/mL using an Amicon 100 kDa molecular weight cut off (MWCO) ultracentrifugal filter (EMD Millipore) pre-equilibrated in buffer A. Protein concentrations were determined photospectromically by absorbance at 280 nm using a theoretical extinction coefficient corresponding to 2 molecules of heterodimeric TarGH ($174125 \, M^{-1} \, cm^{-1}$) calculated using ExPASy software[76]. For downstream assays, *S. aureus* TarGH in buffer A was further purified over a Superose 6 increase 10/300 GL column pre-equilibrated in buffer B (20 mM Tris pH 8, 250 mM NaCl, 0.002% LMNG, 5 mM MgCl$_2$) at 0.2 mL/min in a 4 °C cold-cabinet while samples for cryo-EM were alternatively purified in a high-NaCl buffer C (50 mM Tris pH 8, 500 mM NaCl, 5 mM MgCl$_2$, 0.002% LMNG) (Supplementary Fig. 2). Sample stoichiometry was accessed by SEC-multi-angle light scattering (MALS) on an Agilent 1260 HPLC in-line with a DAWN MALS detector and Optilab differential Refractive index detector (Supplementary Fig. 2). *S. aureus* TarGH samples in buffer C (2 mg/mL, 100 µL) was injected onto a XBridge Premier Protein SEC Column (250 Å, 2.5 µm, 7.8 × 300 mm) at 0.1 mL/min. Background subtraction, peak alignment and subsequent protein conjugate analysis was performed in Astra 8.1.2 software.

## TarGH ATPase activity

The in vitro *S. aureus* TarGH ATPase activity was probed using an established malachite green colorimetric assay[77,78]. ATPase assays were performed in 30 µL reactions shielded from light in a 96 well flat-bottom NUNC tray; gel-filtration purified TarGH was diluted to 2 µM with buffer B supplemented with ATP ranging from 20 mM to 4 µM and incubated at 24 °C for 5 min. After the incubation period, 150 µL of chilled malachite green developing solution (3 M HCl, 1 mg/ml malachite green, 1.5% (w/v) ammonium molybdate and 0.2% (v/v) Tween 20) was added, mixed, and covered with aluminium foil for 10 min at 24 °C. Colorimetric development was quenched by the addition of 75 µL of chilled 34% (w/v) citric acid and the absorbance at 595 nm was immediately analysed in Synergy H4 multimode plate reader (Agilent Technologies). Absorbance measurements corresponding to three isolated experiments were corrected for non-specific ATP hydrolysis against buffer B with ATP. Calculation of inorganic phosphate release was accomplished by direct comparison of absorbance from a K$_2$PO$_4$ standard curve and extrapolated to measured absorbance for each ATP concentration tested. The subsequent mol of inorganic phosphate was divided by 600 s by the initial TarGH protein concentration (2 µM) and plotted using GraphPad Prism 10 with a non-linear regression Michaelis-Menten Least squares fit built-in analysis equation.

Functional inhibition of TarGH ATP-turnover by targocil-II was accomplished through a titration-based ATPase assay modified from above. Purified TarGH in LMNG was incubated with serially diluted targocil-II (Vitascreen LLC) ranging from 100 µM to 2 nM for 10 min on ice before addition of 1 µM ATP. Incubation with ATP, malachite green working solution and colorimetric quenching was performed as above in parallel to TarGH in the absence of targocil-II. Wells corresponding to ATP without TarGH, targocil-II without TarGH and TarGH without ATP and targocil-II were performed as controls to establish baseline ATP-hydrolysis and background inorganic phosphate content. Inhibition assays were repeated in triplicate, translated to µmol using the same K$_2$PO$_4$ standard curved, plotted, fit and statistically analysed in GraphPad Prism 8. Calculation of IC$_{50}$ values were determined using nonlinear regression to a four parameter, variable slope, dose-response-inhibition function in GraphPad.

## Targocil-II−TarGH microscale thermophoresis

The binding affinity for targocil-II for *S. aureus* TarGH was determined using Microscale thermophoresis (MST) with a NanoTemper Technologies Monolith NT.115 instrument. 100 mL of 16 µM TarGH was subjected to cysteine labelling with NT-647-MALEIMIDE (NanoTemper Technologies Labelling Kit RED-MALEIMIDE) according to manufactures

instructions at 3:1 dye to protein ratio in buffer B for 30 min at 24 °C. Labelled TarGH was separated from excess MALEIMIDE dye by desalt fractionation with assay buffer (buffer B (20 mM Tris pH 8, 250 mM NaCl, 0.002% LMNG, 5 mM MgCl$_2$) supplemented with 0.5% (v/v) Tween-20) using kit-provide desalt column (column B). Labelled TarGH was incubated with either non-hydrolysable ATP analogue Adenylyl-imidodiphosphate (AMP-PNP) or Adenosine 5′-O-(3-thio)tri-phosphate (ATPγS) for 15 min at 24 °C. To evaluate binding of unlabelled targocil-II to TarGH, increasing concentrations of targocil-II (6 nM to 200 µM) were used to titrate ATP-bound, MALEIMIDE-labelled TarGH at a constant concentration (5 nM) in premium grade capillaries (NanoTemper technologies). Thermophoresis was measured at 22 °C for 22 s with 10% excitation power, 60% MST power and repeated in at least triplicate. The data were analysed using the NanoTemper MO Affinity analysis software and fitted to a single binding site model based on least squares fit by specific binding with Hill slope equation from GraphPad.

## Gram-positive targocil-II MIC assay

To test the specificity of targocil-II against *S. aureus* amongst other clinically relevant gram-positive pathogens, MIC assays were performed by Dr. Brown at McMaster University in Hamilton, ON, Canada. *S. aureus* (USA300, WCC1135, WCC1139, WCC1133, WCC1149), *S. epidermidis* (ATCC12228), *S. pneumoniae* (WCC1706, WCC1708, WCC1709, WCC1710) and *B. subtilis* (EB6) were grown on solid media overnight (*S. pneumoniae*−blood agar, 37 °C with 5% CO2; *B. subtilis*− cation adjusted Mueller-Hinton agar (CAMHA), 30 °C; all other strains −CAMHA, 37 °C). The following day, colonies were resuspended in saline to an optical density at 600 nm of 0.08−0.1 (O.D$_{600}$). This solution was used to inoculate the appropriate media 1:200. *S. pneumoniae* was tested in round bottom 96-well plates with CAMHB with 3% lysed horse blood (Remel R112478) and incubated at 37 °C with 5% CO2 for 18 h. All other strains were tested in CAMHB in flat bottom 96-well plates and incubated at 30 °C (B. subtilis) or 37 °C (all other strains) for 18 h with final targocil-II concentrations ranging from 64 µg/mL to 0.0625 µg/mL. The final DMSO concentration was 0.5% (v/v) and final volume per well was 100 µL. Assays were repeated in triplicate with threshold for cellular death set at O.D$_{600}$ < 0.04.

## Cryo-EM sample preparation

For all cryo-EM datasets, Quantifoil grids of the 300-mesh variety (R1.2/1.3 or R2/2 in the absence or presence of targocil-II, respectively) were glow discharged for 2 min before 3 µL of protein sample at 2.0−3.0 mg/mL (extinction coefficient corrected 280 nm absorbance) in buffer C was applied to the grid. *E. coli* and *L. lactis* expressed N-terminal tagged *S. aureus* TarGH were incubated for 1 h at 4 °C with 3 mM ATPγS or AMP-PNP at 4 °C whereas *L. lactis* expressed N-terminal tagged *S. aureus* TarGH in the presence of targocil-II was incubated 15 min with 3 mM ATPγS or AMP-PNP and 100 mM targocil-II. All samples applied to grids were blotted for 2−4 s using a Vitrobot Mark IV (Thermo Fisher) at 100% humidity and 4 °C before plunge-freezing into liquid ethane. Grids were transferred to liquid nitrogen and stored until imaging. All grids were screened at the High Resolution Macromolecular Cryo-electron Microscopy (HRMEM) facility (Vancouver, British Columbia, Canada) on a 200 kV Glacios microscope (Thermo Fisher) equipped with a Falcon III camera (Thermo Fisher) using the EPU (Thermo Fisher Scientific) software package. The *S. aureus* TarGH AMP-PNP plus targocil-II dataset was collected at the HRMEM facility at the University of British Columbia using the 300 kV Titan Krios (Thermo Fisher Scientific) transmission cryogenic electron microscope equipped with a Falcon 4i (Thermo Fisher Scientific) direct electron detector and a Selectris (Thermo Fisher Scientific) energy filter. The *S. aureus* TarGH ATPγS dataset was collected at the Pacific Northwest Center for cryo-EM using a 300 kV Titan Krios (Thermo Fisher Scientific) transmission cryogenic electron microscope equipped with the GIF Quantum energy filter (Gatan; operated at 20 keV)

and a K3 Summit (Gatan) direct electron detector. Dataset specific collection parameters (images, pixel size, dose, defocus, range and file format) can be found in Supplementary Table 1.

## Cryo-EM data processing

All processing occurred in CryoSPARC v4.4 as outlined in Supplementary Figs. 3–6 and Supplementary Table 1. Movies were patch motion corrected and patch contrast transfer function (CTF) estimated before blob picking was utilised to generate an initial particle stack. These particles were used to generate reference-free 2D class averages, ab initio reconstructions, and cleaned particle stacks to use in first template based picking and subsequent picking with Topaz[79]. Reference-free 2D classification was used to remove obvious junk particles prior to further processing. Four *ab* initio reconstructions were generated and cleaned through processive heterogenous refinement-based sorting. After careful investigation including C1 refinement, symmetry expansion, symmetry reduction, heterogeneous refinement and 3D classification without alignment, C2 symmetry was applied in subsequent refinements. The final polishing strategy involved successive heterogenous and non-uniform refinements[80]. Final particles underwent reference based motion correction as implemented in CryoSPARC and based on Bayesian polishing[81] and CTF refinements before final reconstruction using either non-uniform refinement or local refinement with a mask around the protein components of the complex.

## Model building and refinement

The ATPγS bound TarGH final map volume was of sufficient quality to permit manual docking of the deposited *Alicyclobacillus herbarius* TarGH structure (PDB: 6JBH). The sequence was adjusted to the *S. aureus* equivalent and the model was rebuilt and refined in Coot[82] and Phenix[83,84]. The resulting model was used for refinement of the other structures using the same approach. Model validation was performed with MolProbity within the Phenix software[85,86].

## Analysis software

Structures were visualised using UCSF ChimeraX[87], UCSF Chimera[88] and PyMOL (Schrödinger). The cavity size of the substrate tunnel was determined using 3V[89] with a default radius probe of 2.0 Å. Interactions were probed using LigPlot[90] and PLIP[91]. Sequence alignments were produced using the ESPript 3.0 server[92]. Sequence conservation was analysed using ConSurf[93]. The positioning of proteins in membranes were predicted using the PPM Web Server[94]. Visual figures were created with GraphPad Prism 10, Chemdraw 22.2.0 and Adobe Illustrator 2024 (Adobe).

## Reporting summary

Further information on research design is available in the Nature Portfolio Reporting Summary linked to this article.

# Data availability

Atomic coordinates have been deposited in the PDB under accession codes 9CFL, 9CFP, 9MHD, 9MHU and 9MHZ. Cryo-EM maps have been deposited in the EMDB under EMD-45550, EMD-45554, EMD-48274, EMD-48281 and EMD-48282. Source data are provided with this paper.

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

## Acknowledgements

We thank Dr. Aleš Berlec of the Jozef Stefan Institute for generously gifting the pNZDual bicistronic plasmid utilised as a template for *L. lactis* cloning presented in this study. We thank Bryan Lin and Helena Sverak for advice on microscale thermophoresis and figure generation, respectively. This work was supported by the Mitacs Accelerate Fellowship (to F.K.K.L.), Michael Smith Health Research BC post-doctoral fellowship award (to T.S.), Tier I Canada Research Chair funding (to N.C.J.S. and E.D.B.) and operating funds from the Canadian Institutes of Health Research (to N.C.J.S. and E.D.B.). We thank Dr. Claire Atkinson, Dr. Florian Rossmann, Joeseph Felt and Dr. Péter Horvath for cryo-EM grid screening and data collection assistance at the High Resolution Macromolecular Electron Microscopy (HRMEM) facility at UBC. We thank the Canadian Foundation of Innovation and British Columbia Knowledge Development Fund (to N.C.J.S.) for infrastructure and operating funds in HRMEM. A portion of this research was supported by NIH grant U24GM129547 and performed at the PNCC at OHSU and accessed through EMSL (grid.436923.9), a DOE Office of Science User Facility sponsored by the Office of Biological and Environmental Research. We thank Dr. Nancy Meyer and Dr. Theo Humphreys for assistance with cryo-EM data collection at PNCC.

## Author contributions

F.K.K.L., S.C.P. and L.J.W. contributed to investigation, methodology, formal analysis, visualisation and writing—original draft, review and editing. M.V., T.S., J.H., A.P. and M.F. contributed to investigation, methodology, formal analysis and writing—original draft, review and editing. N.A.C. contributed to formal analysis, visualisation and writing—review and editing. E.D.B. contributed to methodology, formal analysis, funding acquisition, supervision and writing—review. N.C.J.S. contributed to conceptualisation, methodology, visualisation, formal analysis, funding acquisition, project administration, supervision and writing—original draft, review and editing.

## Competing interests

The authors declare no competing interests.
