## [Transparent Peer Review file · Nature Communications]

Cryo-EM analyses unveil details of mechanism and targocil-II mediated inhibition of *S. aureus* WTA transporter TarGH

Corresponding Author: Professor Natalie Strynadka

Version 0:

Reviewer comments:

Reviewer #1

(Remarks to the Author)

In this manuscript, Franco et al. present five cryo-EM structures of the ABC transporter TarGH from *S. aureus* in different conformations, which they place in the context of the functional cycle, proposing a mechanism for lipid-linked substrate translocation. Within these structures, one is targocil-II bound, additionally providing insights into the transporter inhibition. The structures are of good quality, with one of them even reaching 2.3 Å, which is a particularly high resolution for ABC transporters. Generally, the manuscript is well written, except for the methods part, which requires a bit more polishing. It is undoubtedly an interesting story, which would benefit other researchers in the field, however, I still have some concerns.

Major concerns:

1. The authors rightfully criticize the previously published structure to have flaws in the model building due to moderate resolution (3.9 Å) (lines 79-81). However, they do get the same resolution for a comparable state in their data and confidently assign nucleotide state to ADP there, while claiming that in the previous structure the authors missed ATP (line 81). Can the authors clearly assign whether it is ATP, ADP or a mixture of both at 3.9 Å resolution? While I agree with the authors that ADP is the more likely option, keeping in mind the resolution, the authors should explain their choice. They should also write at which threshold they show the density for ADP in Extended Data Fig 4b and note the thresholds for all other displayed densities, including LMNGE. This applies to all figures with densities. Generally, it is a good idea to show some density for the protein part as well – knowing the threshold for the protein part and for the ligand helps to get an understanding of how much weaker the density for the ligand is.
2. Similarly, building of LMNG molecule at the extracellular site in the 3.9 Å map is borderline supported – there is a poor density there, which is likely LMNG, but since this seems to be the binding site not only for detergents, but also inhibitors, like targocyl-II, and since the authors see endogenously purified nucleotides in that structure, how can they exclude that they also have some copurified compound at the extracellular site? The authors should mention that the assignment of molecules is not trivial at such resolution and that they have to guide their model building based on information from the high-resolution structure.
3. In the loaded state, the authors see additional density for the lipids, which they model with C25, while they write that the only prenyl lipids in *E. coli* are C55. Even though it may be obvious for the structural biologists, it should be explained in the text for the general audience, why the authors build C25 instead of C55 in their structure.
4. All the structures were processed in C2 symmetry since it gave the best resolution, however, C55 binding site seems to be asymmetric. Did the authors look at it in C1? Did they double-check all the ligand densities in the maps without symmetry? How do they look?
5. In Lines 310-311 the authors write: “TarH is bound by an ATP analogue in both cases, suggesting the structural differences are a direct result of targocil-II binding.” However, the authors used different ATP analogs for the two cases: ATPγS for the LOADED state and AMP-PNP for the targocil-II bound state. Therefore, it is not possible to exclude that differences in the D-loop conformation arise from the different analogs present at the active site. If the authors really want to make claims on the allosteric effect of targocil-II on the active site, they would have to collect data with ATPγS and targocil-II, but it would be easier to simply rephrase the text in the manuscript. Especially, since LMNG might also have an effect on the observed structural differences: in the LMNG-free targocil-II bound structure the wedge-shaped pocket, where LMNGC binds, is collapsed without LMNG. Altogether, the authors do not have enough evidence to claim that the differences in the D-loop are “a direct result of targocil-II binding”.

6. The authors performed binding assays with targocil-II only after locking the transporter with AMP-PNP, but does it also bind in the inward open conformation? As a control, they should also perform the targocil-II binding assay in the absence of nucleotide and under conditions where they get an inward open structure.
7. Moreover, I am not convinced about the sequence of how the structures are placed in the functional cycle (Fig. 5). Why would the LOADED structure have to open to the intracellular side before translocation? Is the LOADED structure really representing the functional state of the cycle? Maybe the LOADED structure is simply inhibited by the LMNG molecules – inhibitory effects of detergents have been reported for several different membrane proteins. Did the authors try to get such a LOADED structure with the substrate to prove that it is truly a substrate-binding site where they see LMNGC? If not, they should at least discuss that a DOFF could also be a result of the potential inhibitory effect of LMNGC binding. It is great that the authors propose a mechanism, but it should be clear in the text that this is not the only explanation for the observed structures.

Minor concerns:

1. LMNG is a commonly used detergent and its structure is well known, so panel c should be moved from Figure 1 to the supplements.
2. Line 209: "in" is missing after "observed"
3. Typo in line 649: "2 μ M"
4. Typo in line 670: should be μ L instead of mL – I am sure the authors did not label 100 mL of their protein for MST.
5. Line 686-687: How can TarGH samples be frozen in the absence of TarGH? The sentence should be fixed.
6. The methods section should be polished a bit more in general.

Reviewer #2

(Remarks to the Author)

Summary

TarGH is an essential transporter protein in *Staphylococcus aureus* that is responsible for transporting lipid-linked wall teichoic acid (WTA) precursors from the inner to outer leaflet of the cell membrane. This study used cryo-EM to solve the structures of TarGH in the nucleotide-free, ADP-bound, and ATP-analog with inhibitor targocil-II bound. These structures of TarGH represents five distinct conformations: loaded, inserted, translocated, post-hydrolysis, and reset. The study also captured inhibitor bound state and identified inhibitor binding site. One of the structures ("inserted" state) also captured a C55-P lipid indicating the putative substrate lipid binding site. The authors used these five conformations to propose a lipid-linked substrate translocation mechanism. Overall, the structures reported in this study are good quality and the structural analysis is sound. Many important insights are gleaned from this study. I have several suggestions to address, which I think would help strengthen the manuscript.

Major comments

- In the INSERTED state, the modeled C55-PP exhibits a (PP-cis-trans-trans-trans-cis-cis-trans-cis-trans-trans-) geometry, which does not align with the typical literature-reported configuration, where the first 8 isoprene units are in the cis configuration. Please check and fix the model accordingly. If C55-PP with a proper geometry does not fit the density well, it is possible that the density corresponds to the saturated acyl tail of other type of lipids (fatty acid or phospholipid), not C55-PP. Additionally, the chemical representation of the C55-PP in Figure 1A does not show correct stereochemistry; please correct.
- One of the findings in the study (line 214-215) states that in the LOADED state, the catalytic water is in a non-optimal position for attack due to D-loop in the off conformation. However, the water assignment in the LOADED map does not appear entirely convincing, as the density is elongated and difficult to distinguish from that of water.
- Since LMNG binds at the same site as the inhibitor targocil-II, does the enzyme exhibit [LMNG]-dependent ATPase activity or different levels of activity in the presence of non-maltoside detergent (e.g. digitonin)?
- Although the structural study is well done and the proposed mechanism is feasible, because substrates are partially observed, are inferred from the inhibitor, or modelled without an experimental density, there is still ambiguity of the proposed mechanism. The study lacks sufficient functional data to fully support the proposed transport mechanism. Including mutagenesis data would significantly strengthen the conclusions. For example, the authors can easily mutate a residue on the D loop or a residue that interacts with a phosphate moiety of C55-PP in the inserted state. If functional assays are not feasible, molecular dynamics (MD) simulations could provide valuable insights into the proposed transport mechanism, particularly if TarGH can sample all three states with ATP bound (see below).
- Based on the proposed mechanism, ATP is bound to TarGH during the conformational transition of the loaded, inserted, and translocated states. Given the large movement of NBD (TarH), it is somewhat difficult to envision what triggers this conformational change from the loaded and the inserted states. Please propose possible explanation.
- The authors suggests that Targocil-II binding makes TarGH adopt a catalytically competent conformation. Then how does Targocil-II inhibit TarGH? By blocking the substrate release? If so, its binding affinity seems a bit low. Please clarify the proposed mechanism of Targocil-II inhibition.

Minor comments

- The discussion is too lengthy. I suggest shortening the discussion significantly.
- Please avoid using “convincingly” or “compellingly” and let the readers judge based on the quality of the data.
- Line 183-184 states that LMNG is a structural mimic of C55-P and there are LMNG molecules captured inside the TarGH. Have the authors tried any other detergent? And what is the rationale for 6h LMNG extraction, which seems unusually long for membrane protein extraction.
- Figure 1: The figure contains excessive repetitive labels (e.g., TarG and TarG' are labeled for every structure). Simplifying the labels would improve clarity.
- Figure 2B: The figure includes an extensive list of labeled residues, many of which appear unnecessary. I assume the intention is to highlight positively charged and hydrophobic residues lining the cavity, but other labeled residues seem less relevant. Please consider labeling only the residues critical to the discussion.
- Please clarify what "GM" stands for in the figure 1 legend.
- Figure 3H: The color and text in this figure do not match, and the arrows are too large, obscuring the helices. Consider adjusting the arrows to the side of the helices to better visualize the structural movements.
- In the POST-HYDROLYSIS model (ADP bound), there is no Mg²⁺ in the model. Please check if the density for Mg²⁺ is not present in the map. If not, please comment this point. Additionally, the ADP modeling in this structure appears slightly off. Please review the fitting.

Reviewer #3

(Remarks to the Author)

In this extensive structural work, the authors determined five cryo-EM structures of TarGH, a type V ABC transporter of *Staphylococcus aureus* essential for the export of wall teichoic acids (WTA) that are integral part of the cell wall after being covalently attached to the peptidoglycan layer. The five structures have been determined at highly varying resolutions in the presence of two different ATP analogues, ADP and the inhibitor targocil-II. The targocil-II binding site has been convincingly identified in a 2.9 Å structure. Further, the authors claim to have identified the binding site of a prenyl-lipid (supposedly co-purified from the expression host *E. coli*) at the transporter surface at the middle of the membrane (though, experimental evidence of this finding is questionable, see comments below). Further notable findings were two LMNG binding sites (detergent used to purify TarGH) and a conformational switch of the D-loops that appear to be allosterically coupled to targocil-II binding.

In light of sparse structural information on type V ABC transporters involved in the export of long-chain lipid-linked oligomers (the only system thus far examined was the glycol-lipid exporter WzmWzt from gram-negative *Aquifex aeolicus*), this study on the clinically relevant TarGH protein offers novel insights. The main strength of the paper is the highly resolved (2.3 Å) TarGH structure determined in the presence of ATP-g-S exhibiting a D-loop configuration not compatible with hydrolysis and the targocil-II bound structure (determined in the presence of AMP-PNP) exhibiting a D-loop configuration allowing for ATP hydrolysis. The main weakness of the paper is the complete lack of site-directed mutagenesis data to support structural claims, and a highly speculative (over)interpretation of structural data to propose a mechanism of how WTA is exported. In terms of presentation, the figures are somewhat hard to decipher (though understandably, the matter is complex) and in the text, it is generally difficult to separate “facts from fiction”.

Main points

- 1) The authors observed densities that they interpret to correspond to C25 or C55-PP, but a closer look at the maps indicated that the densities are rather weak and can be interpreted differently. This also applies to the highest resolved 2.3 Å structure (where the authors claim to have located a binding site for C25, which they interpret as the anchor point of the C55 moiety of WTA). In the attached PDF, we made a quick analysis of non-proteinaceous density of the 2.3 Å data showing that at the threshold required to see density for what the authors think is a prenyl chain, there are many equally strong densities (from lipids and/or LMNG) in the vicinity. As a further note, while the side chains in vicinity of the claimed prenyl binding site are very well defined (as expected at 2.3 Å), the quality of density for the lipid is much less defined, suggesting poor occupancy. As a comparison, we also looked at the density of targocil-II (2.9 Å structure), which is clearly defined and beyond any doubt. Unless the authors provide a cryo-EM structure pair with one of them having a prenyl lipid present/added and the other one being for sure devoid of it, it is in our view impossible to unambiguously identify C25/C55-PP binding. Accordingly, the claims the author make regarding the mechanism of WTA export in the paper lack convincing experimental evidence and are thus highly speculative.
- 2) The authors claim that targocil-II allosterically regulates the D-loop conformation, but the cryo-EM structures were determined using ATP-g-S (no targocil-II) or AMP-PNP. The authors “gloss over” the fact that there is a fundamental difference between these two ATP analogues, namely that ATP-g-S is still hydrolysable by ABC transporters, but just much slower than ATP, while AMP-PNP is inert to hydrolysis by ABC transporters. There is ample evidence in the field that the

conformational response of ABC transporters to ATP-g-S versus AMP-PMP can drastically differ. In other words: to really make solid claims about how targocil-II allosterically “governs” D-loop configurations, a structural pair is needed wherein everything is identical (purification, detergent, ATP analogue) except in one case targocil-II is added and in the other case not.

3) There is a logical flaw concerning the targocil-II structure. Namely, the authors claim that targocil-II binding results in full NBD closure and arrangement of the D-loop to allow for ATP hydrolysis. However, they also show (Fig. 3c) that targocil-II inhibits ATP hydrolysis. How does that go together?

4) Much of the claims the authors make are based on structural resemblances of LMNG versus targocil-II versus WTA (Fig. 4a). In our opinion, this interpretation is far-fetched and highly speculative. We suggest the authors make a separate section in the discussion/late results part entitled “speculations on the transport mechanism”, where they can still try to speculate about what they believe to see in the non-proteinaceous densities, but clearly marked and separated from the parts where there is stronger experimental evidence (i.e. D-loop rearrangements and targocil-II binding).

5) The authors mention previous work on WzmWzt of *Aquifex aeolicus*, but they do not even attempt to interpret their data in light of previous work on WzmWzt. While it is certainly the case that much proposed for WzmWzt is speculative, the same also applies for this work. One key mechanistic element shared between TarGH and WzmWzt is the large oligomeric substrate that needs to be threaded across the membrane. In the mechanism proposed in this work (Fig. 6), it remains unsaid/unexplained how the polyol-chain crosses the membrane (step from “inserted” to “translocated”). Would the polymer be threaded through the cavity of TarGH? Or at the TMD surface? Looking at the conformational changes of TarGH from NBD-open to NBD-closed, what conformations would the TMDs assume (inward-facing and outward-facing?). How would the cavities look like in TarGH? Cavity analysis had been conducted for WzmWzt and it would be interesting to compare them to TarGH. And finally, provided the authors succeed in providing solid evidence for the anchor point for the C55 tail is correct (see comment 1), would this anchor point be compatible with a mechanism wherein the two sugars and the polyol tail are threaded through the cavity?

6) The authors do not provide any functional data on structure-inspired site-directed mutants to support their mechanistic claim.

7) The “inserted” (Fig. 1f) structure was made with a differentially tagged version of TarGH, that does not appear to react on ATP-g-S and shows open NBDs and a “weird-looking” inward-facing conformation. Was this TarGH sample tested for ATP hydrolysis activity. And if so, was it active? And can it complement a tarGH knockdown in *S. aureus*? We have doubts whether this structure is functionally relevant.

Minor points

1) Abstract, line 26: No ATP-bound structure is presented in this work, “only” ATP-g-S and AMP-PMP.

2) Line 27: Author statement: “...including a novel conformational role of the D-loop in preventing spurious ATP hydrolysis in the absence of WTA.” We would consider this to be an overinterpretation, because it would imply that the targocil-II bound structure corresponds to the WTA-bound structure.

3) Line 105: does the TarGH sequence of the methicillin-resistant *S. aureus* clinical strain USA300 differ from other *S. aureus* strains? Is there any functional link of TarGH with MRSA, as this statement would imply? If not, we would rather not state it this way.

4) Line 227: say in the main text that you used microscale thermophoresis to determine the targocil-II affinity.

5) In the discussion, pls consider to elude further on the structural/functional role of D-loops that were structurally and functionally studied in other ABC transporters, including BtuCD (PMID: 2300901), TM287/288 (PMID: 25030449), TAP (PMID: 25377891), etc.

6) Please fix the rotamer outliers – in the structures they go up to 10%.

7) Pay attention to the clash score: it should be as low as possible, ideally below 5. Perhaps adding hydrogens during structure refinements could help with clash score and rotamers. Otherwise, we could highly recommend using ISOLDE for model building.

8) Did authors perform refinements with C1 symmetry to confirm that the transporters are indeed C2-symmetrical? We think this is especially critical to check for ligand densities – are they really present in a symmetrical way? For example, an LMNG molecule observed in the extracellular part of TarGH (2.3 Å and 3.9 Å resolution structures). We think it is a bit difficult to interpret such a density located at the C2-symmetry interface. Maybe authors can analyse C1 map in this case to make sure that the observed density corresponds to LMNG molecule?

Reviewer #4

(Remarks to the Author)

Version 1:

Reviewer comments:

Reviewer #1

(Remarks to the Author)

The authors significantly improved the clarity of the manuscript and addressed most of my comments. They even collected

two new datasets, however, the newly obtained structures even strengthened my concerns whether targocil-II is really the direct cause of D-loop switch, since upon targocil binding both conformations with D-loop OFF and D-loop ON can be obtained. In contrast, whenever LMNG-C is bound, only conformation with D-loop OFF is present, as beautifully demonstrated in Fig. S8. Even though the interpretation can be questioned, the data is solid and the observation of the D-loop switch is interesting in itself, so I would be in favour of publishing it and letting the readers judge themselves.

Reviewer #2

(Remarks to the Author)

In this revised manuscript by Strynadka's group, they focused on inhibitor binding and its long-range conformational impact on the D-loop. They also determined TarGH from *L. lactis* and observed similar conformational changes in the D-loop upon targocil-II binding, strengthening their argument for the allosteric regulation between the inhibitor binding site at the extracellular and the cytosolic ATPase at the cytosolic sites. They removed previously reported ADP-bound, apo, and NBD-splayed structures, making the paper much more focused. Although mutagenesis studies on the D-loop would enhance the paper, I believe the revision has significantly improved due to the rearrangement and the new structural data. The inhibitor-bound structures and their proposed role in allosteric communication are important contributions. I recommend the publication of this study.

Reviewer #3

(Remarks to the Author)

We had another careful look at the revised manuscript. The authors did an excellent job in addressing all our concerns, which included the determination of an additional cryo-EM structure of targocil and ATPgS bound TarGH, thereby showing that targocil binding is indeed coupled to the rearrangement of the D-loop. Also the argument that targocil-binding likely results in an overall inhibition of the catalytic cycle makes sense and it is good that this information is now included in the discussion. Most importantly, the paper focusses on those findings which are strongly supported by the data. We therefore recommend to accept the revised paper.

Reviewer #4

(Remarks to the Author)

REVIEWER COMMENTS

We thank the reviewers for the constructive feedback. In considering the comments, we have revised the manuscript to focus largely on the Targocil-II binding and its structural influence on S. aureus TarGH, also comparing it to the structure of the ABC transporter with ATPgS alone to do so. The high resolution of the latter clearly defines a D-loop_{OFF} state, a further important novel mechanistic point of our work and with the targocil-II bound structures in the D-loop_{ON} state showing allosteric regulation between the extracellular bound targocil-II (substrate mimic) and the cytosolic ATPase active sites. These observations and implications are now further supported by added high resolution structures with targocil-II captured in presence of either of ATPgS or AMP-PMP substrate analogs. We have removed the prior described ADP-bound, apo and NBD-splayed structural states for a future paper more focused on the flipping mechanism, which we agree will benefit from added mutational, mass spectrometry and structural experiments that our beyond the scope of this single initial paper. We believe the revised, more focused manuscript has significantly improved readability, figure density and reduced overall speculation.

Reviewer #1 (Remarks to the Author): In this manuscript, Franco et al. present five cryo-EM structures of the ABC transporter TarGH from *S. aureus* in different conformations, which they place in the context of the functional cycle, proposing a mechanism for lipid-linked substrate translocation. Within these structures, one is targocil-II bound, additionally providing insights into the transporter inhibition. The structures are of good quality, with one of them even reaching 2.3 Å, which is a particularly high resolution for ABC transporters. Generally, the manuscript is well written, except for the methods part, which requires a bit more polishing. It is undoubtedly an interesting story, which would benefit other researchers in the field, however, I still have some concerns.

We thank reviewer 1 for their constructive feedback.

Major concerns:

1. The authors rightfully criticize the previously published structure to have flaws in the model building due to moderate resolution (3.9 Å) (lines 79-81). However, they do get the same resolution for a comparable state in their data and confidently assign nucleotide state to ADP there, while claiming that in the previous structure the authors missed ATP (line 81). Can the authors clearly assign whether it is ATP, ADP or a mixture of both at 3.9 Å resolution? While I agree with the authors that ADP is the more likely option, keeping in mind the resolution, the authors should explain their choice. They should also write at which threshold they show the density for ADP in Extended Data Fig 4b and note the thresholds for all other displayed densities, including LMNGE. This applies to all figures with densities. Generally, it is a good idea to show some density for the protein part as well – knowing the threshold for the protein part and for the ligand helps to get an understanding of how much weaker the density for the ligand is.

We agree with the reviewer and did try to exercise caution with interpretation of the nature of the bound nucleotide. We have decided to remove the ADP bound structure from the paper along with the apo and open states and focus on the structures we can be confident in and provide most interesting insights. We have included the map threshold values used for the figures but note these should only be used comparing features within the same map and not between maps.

2. Similarly, building of LMNG molecule at the extracellular site in the 3.9 Å map is borderline supported – there is a poor density there, which is likely LMNG, but since this seems to be the binding site not only for detergents, but also inhibitors, like targocyl-II, and since the authors see endogenously purified nucleotides in that structure, how can they exclude that they also have some copurified compound at the extracellular site? The authors should mention that the assignment of molecules is not trivial at such resolution and that they have to guide their model building based on information from the high-resolution structure.

This 3.9 Å structure has been removed from discussion/paper; See #1

3. In the loaded state, the authors see additional density for the lipids, which they model with C25, while they write that the only prenyl lipids in *E. coli* are C55. Even though it may be obvious for the structural biologists, it should be explained in the text for the general audience, why the authors build C25 instead of C55 in their structure.

We have removed the prenyl lipid from the model. Instead, we have only briefly highlighted the general lipid density observed around the protein and speculated this region specifically could be involved in the sequestering of the lipid-carrier on the surface of TarG (discussion lines 380-392). This aspect will be followed up in a subsequent flippase mechanism paper with additional mass spectrometry, mutagenesis and structural experiments to support and that is beyond the scope of the manuscript here.

4. All the structures were processed in C2 symmetry since it gave the best resolution, however, C55 binding site seems to be asymmetric. Did the authors look at it in C1? Did they double-check all the ligand densities in the maps without symmetry? How do they look?

We did thoroughly investigate if the data had features that significantly broke the symmetry using a variety of approaches (refinement without symmetry, symmetry expansion coupled with 3d classification, symmetry reduction etc) but did not find any significant differences to justify not applying C2 symmetry. A statement in the methods has been added to clarify this (lines 591-594).

5. In Lines 310-311 the authors write: “TarH is bound by an ATP analogue in both cases, suggesting the structural differences are a direct result of targocil-II binding.” However, the

authors used different ATP analogs for the two cases: ATP γ S for the LOADED state and AMP-PNP for the targocil-II bound state. Therefore, it is not possible to exclude that differences in the D-loop conformation arise from the different analogs present at the active site. If the authors really want to make claims on the allosteric effect of targocil-II on the active site, they would have to collect data with ATP γ S and targocil-II, but it would be easier to simply rephrase the text in the manuscript. Especially, since LMNG might also have an effect on the observed structural differences: in the LMNG-free targocil-II bound structure the wedge-shaped pocket, where LMNGC binds, is collapsed without LMNG. Altogether, the authors do not have enough evidence to claim that the differences in the D-loop are “a direct result of targocil-II binding”.

We agree with the reviewer. To address this, we have determined the structure of *L. lactis* produced TarGH (to confirm no difference with *E. coli* produced) with ATP γ S alone and with targocil-II. The ATP γ S alone structure is near identical to the *E. coli* produced protein. With ATP γ S + targocil-II, we see both D-loop_{ON} and D-loop_{OFF} states suggesting that the addition of targocil-II does trigger the conformational change but also that the D-loop is perhaps not as energetically favorable in the presence of ATP γ S compared to AMP-PNP (lines 262-275). This observation is also in keeping with the observed weaker binding of targocil-II with ATP γ S compared to AMP-PNP (Fig. 3c)

6. The authors performed binding assays with targocil-II only after locking the transporter with AMP-PNP, but does it also bind in the inward open conformation? As a control, they should also perform the targocil-II binding assay in the absence of nucleotide and under conditions where they get an inward open structure.

Despite extensive efforts, these binding assays have proved very difficult to perform due to the instability of the heterotetrametric membrane protein in the MST capillaries in absence of analogue (NBDs fall off essentially). However, we have removed any implication from the text that the analogue is required for binding of targocil-II and will aim to develop other methods for a future mechanistic paper that can address these questions.

7. Moreover, I am not convinced about the sequence of how the structures are placed in the functional cycle (Fig. 5). Why would the LOADED structure have to open to the intracellular side before translocation? Is the LOADED structure really representing the functional state of the cycle? Maybe the LOADED structure is simply inhibited by the LMNG molecules – inhibitory effects of detergents have been reported for several different membrane proteins. Did the authors try to get such a LOADED structure with the substrate to prove that it is truly a substrate-binding site where they see LMNGC? If not, they should at least discuss that a DOFF could also be a result of the potential inhibitory effect of LMNGC binding. It is great that the authors propose a mechanism, but it should be clear in the text that this is not the only explanation for the observed structures.

As above, we have revised the manuscript to minimize the flippase mechanistic discussion to reduce speculation around the proposed stages of the export cycle, aspects we will address in a future paper with additional mutagenesis, mass spectrometry and structures with more physiological substrates. We do still hypothesize here that LMNG_c localization in the ATPgS bound structure may be representative of the substrate binding site on the intracellular leaflet prior to translocation. In support, we also briefly mention in the discussion (lines 378-380, Supplementary Fig. 9d) that insertion of the ribitol polymer to initiate translocation could happen via opening of the transporter consistent with the “teepee” state in WzmWzt and also that the exit from the central cavity may occur via the open aromatic seal at the top of the cavity also observed in WzmWzt (Supplementary Fig. 9d). We also point out that the structure of *A. herbarius* TarGH was determined in the presence of DDM and not LMNG and was still in the D-loop_{OFF} state despite no bound DDM in the gate pocket (lines 324-326).

Minor concerns:

1. LMNG is a commonly used detergent and its structure is well known, so panel c should be moved from Figure 1 to the supplements.

Removed.

2. Line 209: “in” is missing after “observed”

3. Typo in line 649: “2 μM”

4. Typo in line 670: should be μL instead of mL – I am sure the authors did not label 100 mL of their protein for MST.

5. Line 686-687: How can TarGH samples be frozen in the absence of TarGH? The sentence should be fixed.

6. The methods section should be polished a bit more in general.

We have updated the Methods section to address points 2-6, thank you for this close reading of our manuscript.

Reviewer #2 (Remarks to the Author):

Summary

TarGH is an essential transporter protein in *Staphylococcus aureus* that is responsible for transporting lipid-linked wall teichoic acid (WTA) precursors from the inner to outer leaflet of the cell membrane. This study used cryo-EM to solve the structures of TarGH in the nucleotide-free, ADP-bound, and ATP-analog with inhibitor targocil-II bound. These structures of TarGH represents five distinct conformations: loaded, inserted, translocated, post-hydrolysis, and reset. The study also captured inhibitor bound state and identified inhibitor binding site. One of the structures (“inserted” state) also captured a C55-P lipid indicating the putative substrate lipid binding site. The authors used these five conformations to propose a lipid-linked substrate translocation mechanism. Overall, the structures reported in this study are good quality and the structural analysis is sound.

Many important insights are gleaned from this study. I have several suggestions to address, which I think would help strengthen the manuscript.

We thank reviewer 2 for their constructive remarks.

Major comments

- In the INSERTED state, the modeled C55-PP exhibits a (PP-cis-trans-trans-trans-cis-cis-trans-cis-trans-trans-) geometry, which does not align with the typical literature-reported configuration, where the first 8 isoprene units are in the cis configuration. Please check and fix the model accordingly. If C55-PP with a proper geometry does not fit the density well, it is possible that the density corresponds to the saturated acyl tail of other type of lipids (fatty acid or phospholipid), not C55-PP. Additionally, the chemical representation of the C55-PP in Figure 1A does not show correct stereochemistry; please correct.

As above, we have removed the INSERTED structure from the manuscript and also the specific modelling of a prenyl lipid in the ATPgS D-loop_{OFF} state structure. We have also updated the stereochemistry in the schematic in Fig. 1a, thank you for this comment. We have noted the presence of lipid like densities (lines 164-167) and left some speculation in the discussion about the specific density under the reentrant helices, suggesting it could provide a conserved site to anchor a C55 prenyl tail from which the substrate linkage unit would be able to access either proposed intracellular or extracellular substrate binding sites (lines 380-392).

- One of the findings in the study (line 214-215) states that in the LOADED state, the catalytic water is in a non-optimal position for attack due to D-loop in the off conformation. However, the water assignment in the LOADED map does not appear entirely convincing, as the density is elongated and difficult to distinguish from that of water.

We agree that the density is elongated but our interpretation of this is due to the non-optimal active site coordination in the D-loop_{OFF} (flipped Val173 carbonyl etc) which is improved in the D-loop_{ON} state. We have tried to make this point more clearly in the text (line 191-193). We note, this density is also similarly present in the D-loop_{OFF} state of the new ATPgS + targocil-II data. Given the potential for different active site conformations in the underlying data we have tried to rule out the density being, for example, from a minor population of D-loop_{ON} but superposition shows the Val173 carbonyl is ~1.9 Å away from this density and feel the best explanation is a water. However, we do not feel the inclusion or exclusion impacts the conclusions so would be happy to omit from the final models and results if the reviewer prefers.

- Since LMNG binds at the same site as the inhibitor targocil-II, does the enzyme exhibit [LMNG]-dependent ATPase activity or different levels of activity in the presence of. non-maltoside detergent (e.g. digitonin)?

This is something we have previously tried to address. Unfortunately, TarGH is not very stable in other detergents. This makes sense in the context of the structure with the specifically bound LMNGs. We do agree LMNG likely competes with targocil-II binding (mitigated somewhat by minimal free detergent in assay conditions) or indeed potential substrate binding in the other LMNG site and this is an ongoing thing we are looking in to. We stress, however, that the ability of targocil-II to inhibit the activity of TarGH is established and not a primary focus of this study.

- Although the structural study is well done and the proposed mechanism is feasible, because substrates are partially observed, are inferred from the inhibitor, or modelled without an experimental density, there is still ambiguity of the proposed mechanism. The study lacks sufficient functional data to fully support the proposed transport mechanism. Including mutagenesis data would significantly strengthen the conclusions. For example, the authors can easily mutate a residue on the D loop or a residue that interacts with a phosphate moiety of C55-PP in the inserted state. If functional assays are not feasible, molecular dynamics (MD) simulations could provide valuable insights into the proposed transport mechanism, particularly if TarGH can sample all three states with ATP bound (see below).

We have taken on board the comments of the speculative nature of some of the structures and the proposed flipping mechanism and have revised the manuscript extensively to focus on the targocil-II and ATPgS structures presented. Although the mechanistic hypothesis is largely removed, we have also tried to stress the speculative nature of remaining hypotheses as appropriate. We agree, and for future studies aimed more at mechanistic aspects of the flippase activity we will pursue mutagenesis, mass spectrometry and capture of native substrates as well as other experiments as suggested by the reviewers.

- Based on the proposed mechanism, ATP is bound to TarGH during the conformational transition of the loaded, inserted, and translocated states. Given the large movement of NBD (TarH), it is somewhat difficult to envision what triggers this conformational change from the loaded and the inserted states. Please propose possible explanation.

We have removed this mechanistic discussion and relevant results from the manuscript until we obtain further data to support these.

- The authors suggests that Targocil-II binding makes TarGH adopt a catalytically competent conformation. Then how does Targocil-II inhibit TarGH? By blocking the substrate release? If so, its binding affinity seems a bit low. Please clarify the proposed mechanism of Targocil-II inhibition.

We have added a section to the discussion to explain how we believe targocil-II is inhibiting the ATPase activity (lines 401-419). In short, we believe targocil-II is

essentially jamming up the transporter and interfering with the conformational changes necessary for post hydrolysis events of the ATPase cycle such as ADP/Pi release or the downstream reset to the start of the cycle. So even if hydrolysis itself still occurred, the slowing down of the other events in the cycle would still lead to the observed inhibition.

Minor comments

- The discussion is too lengthy. I suggest shortening the discussion significantly.

We have extensively updated the manuscript including the discussion and attempted to keep it as succinct as possible.

- Please avoid using “convincingly” or “compellingly” and let the readers judge based on the quality of the data.

We have tried to reduce the use as advised.

- Line 183-184 states that LMNG is a structural mimic of C55-P and there are LMNG molecules captured inside the TarGH. Have the authors tried any other detergent? And what is the rationale for 6h LMNG extraction, which seems unusually long for membrane protein extraction.

As above, unfortunately, TarGH is not very stable in other detergents. This makes sense in the context of the structures with the specifically bound LMNG. We have used a variety of extraction times from 1+ hours and found that 6 hours provides optimal yields.

- Figure 1: The figure contains excessive repetitive labels (e.g., TarG and TarG' are labeled for every structure). Simplifying the labels would improve clarity.

We have updated the figures and legends throughout.

- Figure 2B: The figure includes an extensive list of labeled residues, many of which appear unnecessary. I assume the intention is to highlight positively charged and hydrophobic residues lining the cavity, but other labeled residues seem less relevant. Please consider labeling only the residues critical to the discussion.

We have updated the figures and legends throughout.

- Please clarify what "GM" stands for in the figure 1 legend.

We have updated the figures and legends throughout.

- Figure 3H: The color and text in this figure do not match, and the arrows are too large, obscuring the helices. Consider adjusting the arrows to the side of the helices to better visualize the structural movements.

We have updated the figures and legends throughout.

- In the POST-HYDROLYSIS model (ADP bound), there is no Mg²⁺ in the model. Please check if the density for Mg²⁺ is not present in the map. If not, please comment this point. Additionally, the ADP modeling in this structure appears slightly off. Please review the fitting.

We have removed this structure.

Reviewer #3 (Remarks to the Author):

In this extensive structural work, the authors determined five cryo-EM structures of TarGH, a type V ABC transporter of *Staphylococcus aureus* essential for the export of wall teichoic acids (WTA) that are integral part of the cell wall after being covalently attached to the peptidoglycan layer. The five structures have been determined at highly varying resolutions in the presence of two different ATP analogues, ADP and the inhibitor targocil-II. The targocil-II binding site has been convincingly identified in a 2.9 Å structure. Further, the authors claim to have identified the binding site of a prenyl-lipid (supposedly co-purified from the expression host *E. coli*) at the transporter surface at the middle of the membrane (though, experimental evidence of this finding is questionable, see comments below). Further notable findings were two LMNG binding sites (detergent used to purify TarGH) and a conformational switch of the D-loops that appear to be allosterically coupled to targocil-II binding.

In light of sparse structural information on type V ABC transporters involved in the export of long-chain lipid-linked oligomers (the only system thus far examined was the glycolipid exporter WzmWzt from gram-negative *Aquifex aeolicus*), this study on the clinically relevant TarGH protein offers novel insights. The main strength of the paper is the highly resolved (2.3 Å) TarGH structure determined in the presence of ATP-g-S exhibiting a D-loop configuration not compatible with hydrolysis and the targocil-II bound structure (determined in the presence of AMP-PNP) exhibiting a D-loop configuration allowing for ATP hydrolysis. The main weakness of the paper is the complete lack of site-directed mutagenesis data to support structural claims, and a highly speculative (over)interpretation of structural data to propose a mechanism of how WTA is exported. In terms of presentation, the figures are somewhat hard to decipher (though understandably, the matter is complex) and in the text, it is generally difficult to separate “facts from fiction”.

We thank reviewer 3 for their input.

Main points

1) The authors observed densities that they interpret to correspond to C25 or C55-PP, but a closer look at the maps indicated that the densities are rather weak and can be interpreted differently. This also applies to the highest resolved 2.3 Å structure (where the authors claim to have located a binding site for C25, which they interpret as the anchor point of the C55 moiety of WTA). In the attached PDF, we made a quick analysis of non-proteinaceous density of the 2.3 Å data showing that at the threshold required to see density for what the authors think is a prenyl chain, there are many equally strong densities (from lipids and/or LMNG) in the vicinity. As a further note, while the side chains in vicinity of the claimed prenyl binding site are very well defined (as expected at 2.3 Å), the quality of density for the lipid is much less defined, suggesting poor occupancy. As a comparison, we also looked at the density of targocil-II (2.9 Å structure), which is clearly defined and beyond any doubt. Unless the authors provide a cryo-EM structure pair with one of them having a prenyl lipid present/added and the other one being for sure devoid of it, it is in our view impossible to unambiguously identify C25/C55-PP binding. Accordingly, the claims the author make regarding the mechanism of WTA export in the paper lack convincing experimental evidence and are thus highly speculative.

As above, we now focus on targocil-II induced effects compared to the ATPgS structure and have removed several of the lower resolution structures and specific reference to a prenyl lipid (also removed from the deposited model). Instead, we have now only highlighted the localization of the general lipid like density surrounding the protein and, in the discussion, speculated on the nature of this particular density as a possible conserved site to anchor the substrate carrier lipid (lines 380-392). This will be followed up in a future paper with validated lipids and mutagenesis, but we feel some level of speculation is appropriate in the discussion given the pervasive presence of this lipid like density in many of our structures to date.

2) The authors claim that targocil-II allosterically regulates the D-loop conformation, but the cryo-EM structures were determined using ATP-g-S (no targocil-II) or AMP-PNP. The authors “gloss over” the fact that there is a fundamental difference between these two ATP analogues, namely that ATP-g-S is still hydrolysable by ABC transporters, but just much slower than ATP, while AMP-PNP is inert to hydrolysis by ABC transporters. There is ample evidence in the field that the conformational response of ABC transporters to ATP-g-S versus AMP-PNP can drastically differ. In other words: to really make solid claims about how targocil-II allosterically “governs” D-loop configurations, a structural pair is needed wherein everything is identical (purification, detergent, ATP analogue) except in one case targocil-II is added and in the other case not.

We agree with the reviewer. To strengthen our conclusions, we have determined additional structures with ATPgS alone and with targocil-II added from the same

protein sample purified from *L. lactis*. With ATPgS alone we only observe the D-loop_{OFF} state (like the previous *E. coli* produced protein) whereas after the addition of targocil-II we see both D-loop_{OFF} and D-loop_{ON} (approx. 50% each state). We believe this convincingly demonstrates that targocil-II is inducing the observed allosteric changes; it is interesting that unlike AMP-PNP, with ATPgS we see less than full occupancy of the D-loop_{ON}, suggesting that with the latter this state is less energetically favourable. This observation is supported by the binding data we present for both ATP analogues (Fig. 3c). We have included these structures in the updated manuscript.

3) There is a logical flaw concerning the targocil-II structure. Namely, the authors claim that targocil-II binding results in full NBD closure and arrangement of the D-loop to allow for ATP hydrolysis. However, they also show (Fig. 3c) that targocil-II inhibits ATP hydrolysis. How does that go together?

The cycle of ATP hydrolysis involves not only ATP->ADP but also Pi release and reset. Given we are measuring Pi as a marker for hydrolysis, anything that interferes with the cycle will show a reduction of activity. Targocil-II binding could equally be interfering with the conformational changes necessary for these other steps, essentially jamming up the transporter, even if the hydrolysis event itself still happens. We have updated the discussion to try and make this point clearer (lines 401-419).

4) Much of the claims the authors make are based on structural resemblances of LMNG versus targocil-II versus WTA (Fig. 4a). In our opinion, this interpretation is far-fetched and highly speculative. We suggest the authors make a separate section in the discussion/late results part entitled “speculations on the transport mechanism”, where they can still try to speculate about what they believe to see in the non-proteinaceous densities, but clearly marked and separated from the parts where there is stronger experimental evidence (i.e. D-loop rearrangements and targocil-II binding).

As above, in updating the manuscript we have moved away from the more speculative nature of the flippase mechanism and have omitted several of the structures that contributed to this. However, we do not believe that the idea that the observed binding of LMNG (especially to the gating pocket previously proposed to be a substrate binding site for WzmWzt) or targocil-II (with similar physicochemical properties to the substrate linkage unit and ability to induce the conformational changes we observe) could be informative for substrate binding is far-fetched. Indeed, many small molecule inhibitors bind as substrate mimetics eg beta-lactams, and vice versa, substrate complexes are used as starting points for in silico drug design. We have, however, tried to stress the more speculative nature of these proposal as appropriate throughout.

5) The authors mention previous work on WzmWzt of *Aquifex aeolicus*, but they do not even attempt to interpret their data in light of previous work on WzmWzt. While it is

certainly the case that much proposed for WzmWzt is speculative, the same also applies for this work. One key mechanistic element shared between TarGH and WzmWzt is the large oligomeric substrate that needs to be threaded across the membrane. In the mechanism proposed in this work (Fig. 6), it remains unsaid/unexplained how the polyol-chain crosses the membrane (step from “inserted” to “translocated”). Would the polymer be threaded through the cavity of TarGH? Or at the TMD surface? Looking at the conformational changes of TarGH from NBD-open to NBD-closed, what conformations would the TMDs assume (inward-facing and outward-facing?). How would the cavities look like in TarGH? Cavity analysis had been conducted for WzmWzt and it would be interesting to compare them to TarGH. And finally, provided the authors succeed in providing solid evidence for the anchor point for the C55 tail is correct (see comment 1), would this anchor point be compatible with a mechanism wherein the two sugars and the polyol tail are threaded through the cavity?

As above, we have removed these structures and the more speculative mechanistic proposals. We have included (and left in) discussion of the previous WzmWzt structures, especially the “teepee” conformation and structure with an open aromatic seal to create an extracellular channel when discussing the potential route of translocation (lines 378-380; Supplementary Fig. 9). We also suggest how the lipid linkage unit could pass the membrane with the hydrophilic polymer through the cavity and the hydrophobic tail sequestered on the outside surface (line 388-392). This is consistent to what has been proposed for WzmWzt as stated on line 392. We have also included a cavity analysis comparing the D-loop_{OFF} and D-loop_{ON} states (Fig. 4a).

6) The authors do not provide any functional data on structure-inspired site-directed mutants to support their mechanistic claim.

We have removed and toned down much of the flippase mechanistic claims and will add appropriate experiments for a future paper.

7) The “inserted” (Fig. 1f) structure was made with a differentially tagged version of TarGH, that does not appear to react on ATP-g-S and shows open NBDs and a “weird-looking” inward-facing conformation. Was this TarGH sample tested for ATP hydrolysis activity. And if so, was it active? And can it complement a tarGH knockdown in *S. aureus*? We have doubts whether this structure is functionally relevant.

We have removed this structure from the manuscript.

Minor points

1) Abstract, line 26: No ATP-bound structure is presented in this work, “only” ATP-g-S and AMP-PMP.

We have updated the descriptions of the states to better reflect what has been captured.

2) Line 27: Author statement: "...including a novel conformational role of the D-loop in preventing spurious ATP hydrolysis in the absence of WTA." We would consider this to be an overinterpretation, because it would imply that the targocil-II bound structure corresponds to the WTA-bound structure.

We have tried to stress throughout that these are speculations.

3) Line 105: does the TarGH sequence of the methicillin-resistant *S. aureus* clinical strain USA300 differ from other *S. aureus* strains? Is there any functional link of TarGH with MRSA, as this statement would imply? If not, we would rather not state it this way.

We have genomic DNA for USA300 and this is what we used as a template for making these recombinant expression plasmids. The deposited pDBs are also annotated with these UniProt sequences. It is not uncommon for orthologues in different strains to have small variations in sequence. Further, there is a functional link in that inhibition with targocil-II sensitizes MRSA to beta-lactams.

4) Line 227: say in the main text that you used microscale thermophoresis to determine the targocil-II affinity.

We have added this in.

5) In the discussion, pls consider to elude further on the structural/functional role of D-loops that were structurally and functionally studied in other ABC transporters, including BtuCD (PMID: 23000901), TM287/288 (PMID: 25030449), TAP (PMID: 25377891), etc.

We thank the reviewer for these citations and have updated the discussion to include these (lines 339-350). We have tried to be as thorough as possible in looking for other structures/studies into the role of the D-loop. However, we have not come across any with such large conformational changes in the D-loop as we observe here. We hope our observations here provide further information to establish the structural events involved in this interesting allostery.

6) Please fix the rotamer outliers – in the structures they go up to 10%.

Updated.

7) Pay attention to the clash score: it should be as low as possible, ideally below 5. Perhaps adding hydrogens during structure refinements could help with clash score and rotamers. Otherwise, we could highly recommend using ISOLDE for model building.

Updated.

8) Did authors perform refinements with C1 symmetry to confirm that the transporters are indeed C2-symmetrical? We think this is especially critical to check for ligand densities – are they really present in a symmetrical way? For example, an LMNG molecule observed in the extracellular part of TarGH (2.3 Å and 3.9 Å resolution structures). We think it is a bit difficult to interpret such a density located at the C2-symmetry interface. Maybe authors can analyse C1 map in this case to make sure that the observed density corresponds to LMNG molecule?

As above, we performed extensive refinements in C1 (symmetry expansion, reduction, 3D classification etc) prior to being convinced of the justification of applying C2 symmetry. Even for the LMNG_E, there was never any justification of maintaining C1, indeed given the low signal of this molecule the correct “true” alignment of a complex where this was the only place that the symmetry was broken in a non-random way would be very hard to detect and classify. We have added a statement to the Methods to reflect the handling of symmetry (lines 591-593).

Reviewer #4 (Remarks to the Author):

We thank reviewer 4 for their input.

We thank for the reviewers for their constructive feedback throughout the review process.

REVIEWERS' COMMENTS

Reviewer #1 (Remarks to the Author):

The authors significantly improved the clarity of the manuscript and addressed most of my comments. They even collected two new datasets, however, the newly obtained structures even strengthened my concerns whether targocil-II is really the direct cause of D-loop switch, since upon targocil binding both conformations with D-loop OFF and D-loop ON can be obtained. In contrast, whenever LMNG-C is bound, only conformation with D-loop OFF is present, as beautifully demonstrated in Fig. S8. Even though the interpretation can be questioned, the data is solid and the observation of the D-loop switch is interesting in itself, so I would be in favour of publishing it and letting the readers judge themselves.

Reviewer #2 (Remarks to the Author):

In this revised manuscript by Strynadka's group, they focused on inhibitor binding and its long-range conformational impact on the D-loop. They also determined TarGH from *L. lactis* and observed similar conformational changes in the D-loop upon targocil-II binding, strengthening their argument for the allosteric regulation between the inhibitor binding site at the extracellular and the cytosolic ATPase at the cytosolic sites. They removed previously reported ADP-bound, apo, and NBD-splayed structures, making the paper much more focused. Although mutagenesis studies on the D-loop would enhance the paper, I believe the revision has significantly improved due to the rearrangement and the new structural data. The inhibitor-bound structures and their proposed role in allosteric communication are important contributions. I recommend the publication of this study.

Reviewer #3 (Remarks to the Author):

We had another careful look at the revised manuscript. The authors did an excellent job in addressing all our concerns, which included the determination of an additional cryo-EM structure of targocil and ATPgS bound TarGH, thereby showing that targocil binding is indeed coupled to the rearrangement of the D-loop. Also the argument that targocil-binding likely results in an overall inhibition of the catalytic cycle makes sense and it is good that this information is now included in the discussion. Most importantly, the paper focusses on those findings which are strongly supported by the data. We therefore recommend to accept the revised paper.

Reviewer #4 (Remarks to the Author):

Reviewer #3 attachment:

“Loaded” 2.3 Å. Protein densities in blue, C25 density in orange, other non-proteinaceous density in grey.

Same as above, but at lower density threshold to cover entire C25 lipid tail.

“Translocated”, 2.9 Å. Protein in green and non-proteinaceous density for targocil-II in orange. Other non-proteinaceous densities would show in grey, but they are not visible at this threshold.